# A multiparametric anti-aging CRISPR screen uncovers a role for BAF in protein synthesis regulation

Sophia Y. Breusegem[1,6], Jack Houghton[1,7], Raquel Romero-Bueno[2,10], Adrián Fragoso-Luna [2,10], Katherine A. Kentistou [3], Ken K. Ong [3], Anne F. J. Janssen[1,8], Nicholas A. Bright [1], Christian G. Riedel [4], John R. B. Perry [3,5], Peter Askjaer [2] & Delphine Larrieu [1,9] ✉

Progeria syndromes are very rare, incurable premature aging conditions recapitulating most aging features. Here, we report a whole genome, multiparametric CRISPR screen, identifying 43 genes that can rescue multiple cellular phenotypes associated with progeria. We implement the screen in fibroblasts from Néstor-Guillermo Progeria Syndrome male patients, carrying a homozygous A12T mutation in BAF. The hits are enriched for genes involved in protein synthesis, protein and RNA transport and osteoclast formation and are validated in a whole-organism *Caenorhabditis elegans* model. We further confirm that BAF A12T can disrupt protein synthesis rate and fidelity, which could contribute to premature aging in patients. This work highlights the power of multiparametric genome-wide suppressor screens to identify genes enhancing cellular resilience in premature aging and provide insights into the biology underlying progeria-associated cellular dysfunction.

Premature aging syndromes (progerias) are very rare conditions that recapitulate many aspects of physiological aging, causing symptoms such as alopecia, lipodystrophy, cardiovascular dysfunction, and bone dysfunction well before the expected age of onset. Many progeria syndromes are caused by mutations in nuclear envelope (NE) associated proteins. For example, the classic Hutchinson-Gilford progeria syndrome (HGPS) is caused by mutations in *LMNA* encoding for the nuclear lamina proteins lamin A and C[1,2]. A more recently described progeria, Néstor-Guillermo progeria syndrome (NGPS), is caused by a recessive alanine to threonine amino acid substitution at position 12

(p.Ala12Thr) in the *BANF1* gene, encoding the 10 KDa protein barrier-to-autointegration factor (BAF)[3,4]. BAF forms dimers that bind to DNA[5,6], lamin proteins as well as LEM-domain containing proteins of the inner nuclear membrane including emerin[7,8]. Through these interactions, BAF exerts critical functions including reformation of the NE after mitosis[9,10] and NE rupture repair[11]. NGPS patients suffer from multiple aging-associated pathologies, such as alopecia and lipodystrophy, as well as osteoarthritis and joint stiffness. However, the most severe phenotype affecting their quality of life is bone dysfunction, with osteoporosis and dramatic skeletal deformation[3,4]. NGPS patients have an increased

[1]Cambridge Institute for Medical Research, University of Cambridge, Cambridge Biomedical Campus, Keith Peters Building, Hills Road, Cambridge, UK. [2]Centro Andaluz de Biología del Desarrollo (CABD), Consejo Superior de Investigaciones Científicas-Universidad Pablo de Olavide-Junta de Andalucía, Seville, Spain. [3]MRC Epidemiology Unit, University of Cambridge School of Clinical Medicine, Institute of Metabolic Science, Cambridge, UK. [4]Karolinska Institutet, Blickagången 16, 141 57, Huddinge, Sweden. [5]Metabolic Research Laboratory, Wellcome-MRC Institute of Metabolic Science, University of Cambridge School of Clinical Medicine, Cambridge, UK. [6]Present address: Sophia Y. Breusegem: MRC toxicology unit, University of Cambridge, Tennis Court Road, Cambridge, UK. [7]Present address: Jack Houghton: Imperial College London, Exhibition Road, South Kensington, London, UK. [8]Present address: Anne F. J. Janssen: Institute for Molecules and Materials, Radboud University, Heyendaalseweg 135, Nijmegen, The Netherlands. [9]Present address: Delphine Larrieu: Altos Labs, Cambridge Institute of Science, Cambridge, UK. [10]These authors contributed equally: Raquel Romero-Bueno, Adrián Fragoso-Luna. ✉e-mail: dlarrieu@altoslabs.com

life expectancy compared to HGPS, probably due to the absence of cardiovascular dysfunction, the main cause of death in HGPS[12,13].

NGPS remains poorly characterized compared to HGPS, and insights into the molecular mechanisms behind the disease have only recently started to emerge[14–16]. Therefore, with the aim of identifying genes and pathways relevant to NGPS, as well as potential targets for therapeutic avenues in the disease, we carried out a whole genome CRISPR/Cas9 arrayed microscopy screen in patient-derived NGPS cells. We assessed the impact of deleting each of the ~20,000 human genes on four specific cellular NGPS phenotypes simultaneously, to identify "rescue" genes acting across several phenotypes. Through this approach, we identified 43 genes enriched in pathways including protein synthesis and bone cell development, that improve NGPS cellular phenotypes.

## Results

### NGPS fibroblasts show distinct phenotypes compared to HGPS

Fibroblasts from HGPS patients have been extensively studied and characterized. The best-established phenotypes in these cells include NE blebs and invaginations, loss of heterochromatin, downregulation of lamin B1, nucleocytoplasmic transport defects and accumulation of DNA damage foci[17–22]. To establish whether NGPS cells might share similar phenotypes, we obtained immortalized fibroblasts from two of the three so far identified NGPS patients (NGPS1 and NGPS2—both males) and age-matched immortalized fibroblasts (wild-type—WT) (gift from Dr. Carlos Lopez-Otin). These are the only NGPS cells available to the scientific community, as non-immortalized primary cells could not be maintained in culture (personal communication from Dr. Carlos Lopez-Otin). We observed severe NE abnormalities by electron microscopy, including deep folds and NE blebbing but no apparent loss of heterochromatin at the nuclear periphery (Fig. 1A). The immortalized NGPS cells proliferated at a rate comparable to WT cells. As previously reported[4,23], we confirmed the accumulation of the NE protein emerin into the cytoplasm of both NGPS1 and 2 (Fig. 1B), and the absence of lamin B1 downregulation (Fig. 1B) and of DNA double strand breaks accumulation (53BP1 foci Fig. 1C, D), both of which being common hallmarks of HGPS cells. In addition, and unlike HGPS cells, NGPS fibroblasts did not display loss of nuclear Ran—used as a reporter for nucleocytoplasmic transport (Fig. 1C, E) or of the heterochromatin markers HP1γ (Fig. 1F–I), H3K9me3 (Fig. 1J and Supplementary Fig. 1A–C) or H3K79me2 (Fig. 1K). We did however observe an increase in the expression of the cyclin-dependent kinase inhibitor p21 – involved in cell cycle progression, apoptosis and DNA damage response—in both NGPS cell lines but to different extents (Fig. 1L and Supplementary Fig. 1D, E).

### A combination of four specific NGPS phenotypes amenable to high-throughput screening

Phenotypes associated with NE dysfunction in progeria cells can only be assessed by microscopy, precluding a whole genome pooled CRISPR screening approach. We therefore designed a multiparametric, microscopy based CRISPR screen, aimed at interrogating the whole genome to identify genes that when deleted, could rescue multiple NGPS cellular phenotypes. This approach is based on the principle of synthetic rescue, relying on a genetic interaction whereby a protein not involved in the etiology of the disease is targeted, thereby rebalancing the pathophysiological state towards a "healthy" one. Our previous work, based on small scale screening, has established the proof-of-concept of this approach in HGPS, where we identified N-acetyltransferase 10 (NAT10) as a target to reverse HGPS aging phenotypes[24–26]. The design of the screen is depicted in Fig. 2A and involves multiple steps. The first major step relied on the identification

of a combination of phenotypes, specific to the BAF A12T mutation, significantly different in NGPS cells compared to WT cells, and amenable to high throughput screening. To this aim, we took advantage of our recently established NGPS2 isogenic cell lines, in which we reversed the homozygous BAF A12T mutation using CRISPR/Cas9 (NGPS2 corrected)[14]. For this reason, we carried out further validation using the NGPS2 cell line. The screen set-up also required the design of specific pipelines to reliably identify and quantify these phenotypes. The first of these phenotypes is the delocalization of emerin from the nucleus into the cytoplasm (Fig. 2B–C and Supplementary Fig. 2A), as characterized by previous studies[4,23]. The second phenotype is the increased nuclear deformation, commonly observed in progeria cells[17,27] and in cells from normal aged individuals. This was measured using a perimeter to area ratio (P2A) (Fig. 2D and Supplementary Fig. 2B). Additionally, we identified an increased number of NE ruptures during interphase in NGPS cells (Fig. 2E and Supplementary Fig. 2C), further characterized in our recently published study[14]. In that study, we showed that the BAF A12T mutation in NGPS cells affects the binding affinity between BAF and Lamin A/C, preventing the recruitment of Lamin A/C to the sites of NE ruptures, leading to frequent NE re-rupturing in interphase cells. Based on these findings, here we identified NE ruptures on fixed cells by the presence of nuclear blebs (observed using LAP2 staining), the first step in the NE rupturing process[28–30] (Fig. 2E, F, H). Finally, we observed an increased frequency of micronuclei formation in NGPS cells (Fig. 2G, I and Supplementary Fig. 2D), a known marker of genomic instability. The combination of these four phenotypes was subsequently chosen for the whole genome screen.

Next, we engineered stable Cas9 expression in WT and NGPS fibroblasts for use in the CRISPR screening. After testing the efficiency of protein transduction using various promoters (Supplementary Fig. 3A, B) we infected WT and NGPS cells with hCMV-Cas9. Despite a good Cas9 expression level in all three cell lines (Supplementary Fig. 3C), the knock-out efficiency in individual cells was too heterogenous for screening purposes (Supplementary Fig. 3D). Therefore, we established individual WT and NGPS Cas9 expressing clones grown from single cells and selected the best clones based on Cas9 efficiency assessed by resistance to 6-Thioguanine after HPRT knock-out[31] (Supplementary Fig. 3E–G). We also assessed Cas9 expression level by western blotting (Supplementary Fig. 3E) and picked clones combining a high cutting efficiency, and no increase in the level of the DNA double strand break marker γH2AX (Supplementary Fig. 3E, F). Finally, to avoid obtaining results that could result from a clonal effect, we pooled 3 of the selected clones to generate the WT-Cas9, NGPS1-Cas9 or NGPS2-Cas9 cell populations. The NGPS2-Cas9 was the one used for the screen and was generated by pooling clones D, F and I (Supplementary Fig. 3E-G). These Cas9 cells showed a good level of protein knock-down upon transient transfection with various crRNAs (Fig. 2J— acetyl-tubulin level serves as an additional readout for HDAC6 knock-out efficiency).

Finally, as the project started at a time when the first whole genome CRISPR arrayed libraries were being synthetized, we tested the efficiency of the two non-viral systems available at the time, one relying on a CRISPR/tracrRNA (cr/tracrRNA), the other one on a single guide RNA (sgRNA). After comparing the knock-down efficiency of several genes by western blotting (Supplementary Fig. 3H) and by immunofluorescence (Fig. 2K and Supplementary Fig. 3I–L), we set up a mini screen relying on cellular proliferation arrest upon knock-down of five different mitotic genes, compared to 5 non-mitotic genes (Fig. 2L). In both cases, we observed a strong proliferation arrest 3 days after transfection of either the sgRNA (Fig. 2L left panel) or the cr/tracrRNA (Fig. 2L right panel) against the mitotic genes (blue/purple genes), without transfection-associated toxicity (gray/red/yellow genes). Based on cost efficiency, we therefore selected the cr/tracrRNA system for the whole genome screen.

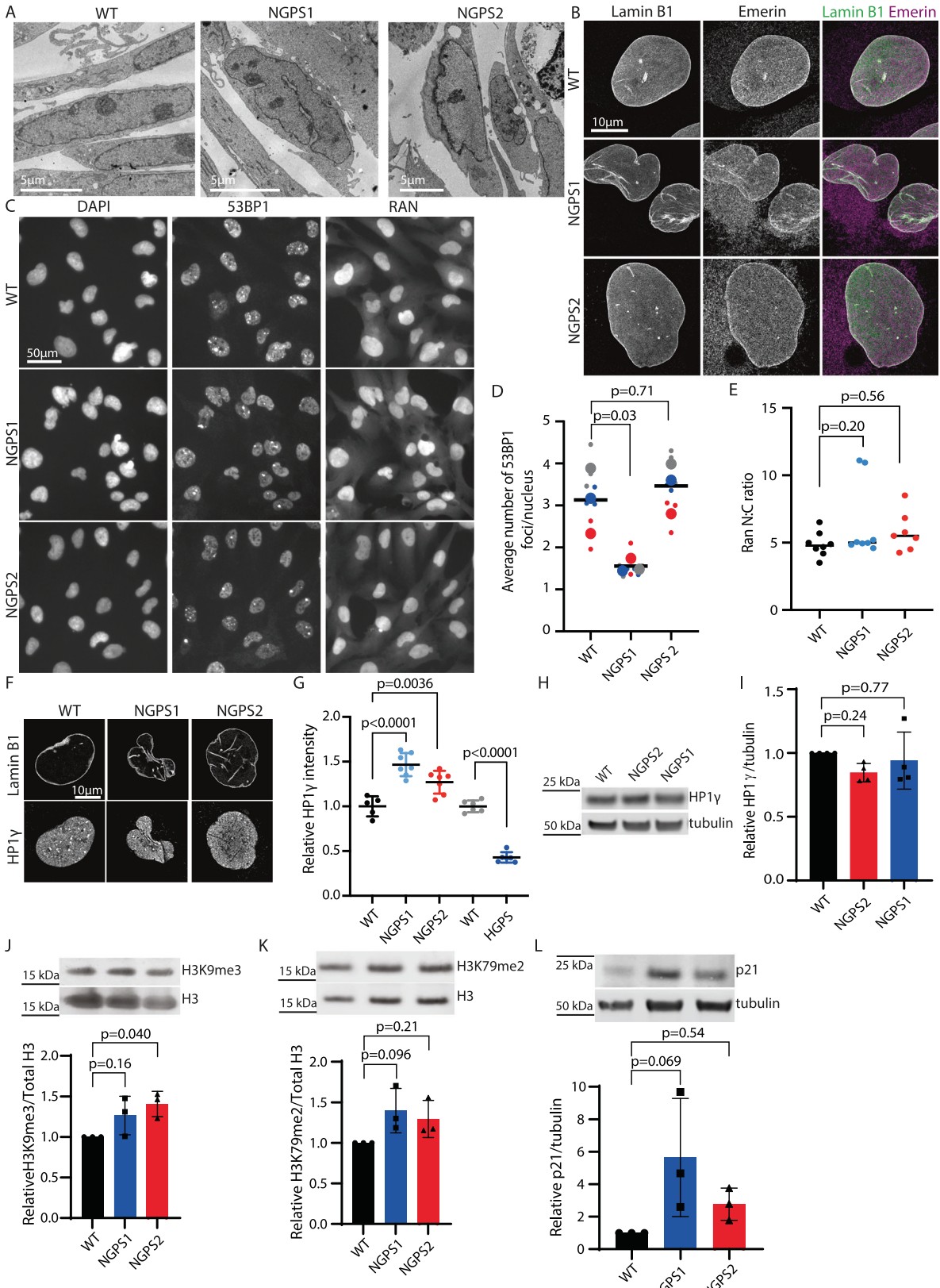

## The multiparametric NGPS screen identifies 43 hits that reverse NGPS phenotypes

We then carried out the primary genome wide screen using the pool of 3 NGPS2-Cas9 expressing clones (NGPS2-Cas9). An overview of the screening approach is depicted in Fig. 3A, and the plate layouts are shown in Supplementary Fig. 4A–B. The four screening phenotypes—

nuclear shape, micronuclei, NE ruptures (blebs) frequency and emerin nuclear intensity - were reduced to two dimensions for each single knock-out and mapped as a Uniform Manifold Approximation and Projection (UMAP). The cluster analysis (Fig. 3B) was used to visualize NGPS2 cells (gray dots) alongside the same parameters measured in control cells (blue dots). The two populations were clearly separated,

**Fig. 1 | NGPS fibroblasts show distinctive phenotypes to HGPS. A** Representative transmission electron micrographs of WT, NGPS1 and NGPS2 fibroblasts (males). **B** Representative high resolution microscopy images showing the expression and localization of lamin B1 and emerin in WT and NGPS fibroblasts. **C** Immunofluorescence images showing DNA damage foci (53BP1) and nucleocytoplasmic transport (Ran) in WT and NGPS fibroblasts. Images were obtained with the CX7 high-content microscope and quantified in (**D**) and (**E**) using the HCS Studio™ software. **D** Superplot of the data (3 technical replicates in 3 independent experiments), lines indicate average values, statistical analysis using nested one-way ANOVA with Dunnett's multiple comparisons. **E** Data from 3 independent experiments, 2-3 technical replicates each, lines indicate average values, statistical comparison using one-way ANOVA with Dunnett's multiple comparisons. **F** High

resolution microscopy images of the heterochromatin marker HP1γ in WT and NGPS cells. **G** Nuclear HP1γ levels quantified using the high-content microscope in 2 independent experiments, each averaging 500 nuclei in 3 wells. Means ± SD are shown. For the NGPS cells statistical testing used one-way ANOVA with Dunnett's multiple comparisons; HGPS cells were compared to WT using two-tailed unpaired t-test. **H** Representative Western blot and (**I**) quantitative analysis of 4 independent experiments showing HP1γ in NGPS cells compared to WT. **J–L** Representative western blots and quantitative analysis of 3 independent experiments, showing the heterochromatin marks H3K9me3 (**J**) and H3K79me2 (**K**) or the aging and senescence marker p21 (**L**) in NGPS cells compared to WT. In **I–L** data points are overlaid on columns indicating the mean +/- SD, and statistical analysis used one-way ANOVA with Dunnett's multiple comparisons.

reflecting the phenotypic difference between the control and the NGPS2 cells. The red dots highlight 112 genes that upon being depleted were identified as "normalizing" the combination of NGPS phenotypes. To these, we added 18 genes not yet on the list that appeared in the top 15 of three individual phenotypes (see details in the "Methods" section). The top 20 genes improving or worsening each individual phenotype in NGPS cells (nuclear shape, nuclear Emerin, micronuclei and nuclear bleb frequency) is presented in Fig. 3C–F.

We then carried these 130 genes into a validation screen, in which 3 out of the 4 individual crRNAs used in the primary screen were deconvoluted into individual wells (Fig. 3A). We carried out the validation screen in triplicate and confirmed 43 hits with high confidence (at least 2 out of 3 crRNA and confirmed in the 3 replicates). A gene ontology analysis revealed specific enrichment in biological processes including translation, protein and RNA transport as well as osteoclast development−of high relevance to the patient phenotypes (Fig. 4A, B). In accordance with the concept of synthetic rescue, only 3 hits were proteins associated with the NE: NUP160, SEC13 and AHCTF1 (ELYS). These genes were able to improve multiple nuclear envelope phenotypes at once, as observed in representative immunofluorescence images from the validation screen (Fig. 4C).

To evaluate the potential relevance of our identified candidate genes to normal variation in related health and disease traits, we interrogated phenotypic and genetic data from available large-scale population studies. These analyzes aim to assess the impact of naturally occurring alleles, influencing the function of our candidate genes, on related phenotypes. Since NGPS patients display severe skeletal abnormalities and lipoatrophy, we focused on publicly available Genome Wide Association Studies (GWAS) of body mass index (BMI), waist-hip ratio (WHR) adjusted for BMI, circulating triglyceride levels and estimated bone mineral density (eBMD) in sample sizes up to 1,253,275 individuals (Fig. 4D). We found that 30 of our 43 identified genes were within 500 kb of a genome-wide significant signal for at least one of these four traits. Specifically, 13 genes were proximal to BMI signals, 11 to WHR, 19 to eBMD and 13 to triglyceride signals (Supplementary Data 8 & Fig. 4D). To more directly link these proximal associated genetic variants to the function of our candidate genes, we undertook a number of variants to gene mapping approaches (see "methods"), including assessment of protein-coding variants and integrating activity-by-contact enhancer maps and expression quantitative trait loci (eQTL) data. These analyzes highlighted a number of genes with strong support for a direct involvement in these human phenotypes (Fig. 4D). For example, genetic variants residing within enhancers for *RPL13* and influencing its transcript expression in blood, were associated with eBMD, triglycerides and BMI. Collectively these findings demonstrate that our cellular screens are able to identify genes that influence NPGS-like phenotypes in the normal population.

## Increased protein synthesis rate associated with BAF A12T is reduced upon depletion of the hits

The screen revealed a strong enrichment for genes involved in protein synthesis, whose modulation has previously been associated with longevity in various model organisms. To gain further insights into protein synthesis regulation in NGPS, we first carried out RNA-Seq analysis in wild-type cells compared to NGPS1 and NGPS2. The analysis revealed 213 genes as being differentially expressed in the patient cell lines compared to control. Gene ontology analysis identified a strong enrichment for RNA processing and translation regulation (Fig. 5A–B). To assess how this could impact translation rate, we measured nascent protein synthesis using incorporation of the clickable methionine analog L-homopropargylglycine (HPG). Upon fluorescent labeling of HPG through a "click" reaction, we observed a significant increase in protein synthesis in NGPS cells compared to both WT cells and NGPS corrected cells (Fig. 5C, Supplementary Fig. 5A) as well as in wild-type fibroblasts overexpressing BAF A12T compared to BAF WT (Fig. 5D, Supplementary Fig. 5B). This suggests that the BAF A12T mutation can drive increased protein synthesis, similarly to what has been reported previously in HGPS cells[32]. However, even though we observed a modest but significant increase in the nucleolar area of NGPS2 cells compared to NGPS2 corrected cells (Fig. 5E–F), as well as in fibroblasts expressing BAF A12T compared to BAF WT (Fig. 5G), no significant difference was observed between the WT cells and the NGPS cells (Supplementary Fig. 5C, D). Interestingly, alongside this phenotype, we identified through a dual luciferase mistranslation assay[33] that both NGPS1 and NGPS2 cells displayed an increased rate of errors during protein synthesis, as observed by an increased readthrough (Fig. 5H, Supplementary Fig. 5E).

We next asked how depletion of the hits identified in the screen might impact nascent protein synthesis in NGPS cells. To address this question, we used siRNA to deplete 41 of the hit proteins, followed by HPG incorporation and quantification of HGP fluorescent intensity by high-throughput microscopy (Fig. 6A, Supplementary Fig. 5F). This confirmed a higher rate of protein synthesis in NGPS2 cells (Fig. 6A red dotted line) compared to NGPS2 corrected cells ("NGPS WT"). As expected, depletion of the hits directly involved in protein synthesis (*RPL12, RPL37A, RPL13, RPS3A, EIF3B*) almost completely abolished HPG incorporation, and served as a good positive control for the assay. Interestingly however, we observed that depletion of many of the other hits, not known to play a direct role in protein synthesis, also led to a reduction of HPG incorporation to various extents in NGPS2 cells, with 21 genes out of the 43 hits (49%) reaching significance (Fig. 6A, blue data points). To assess the potential link between the reduction of protein synthesis we observed in NGPS2 upon depletion of the hits and the phenotypic improvements identified in our whole genome screen, we treated NGPS1 and 2 cells with the protein synthesis inhibitors cycloheximide (CHX) and silvestrol. Both inhibitors led to the enrichment of emerin into the nucleus of NGPS1 and NGPS2 cell lines (Fig. 6B−C).

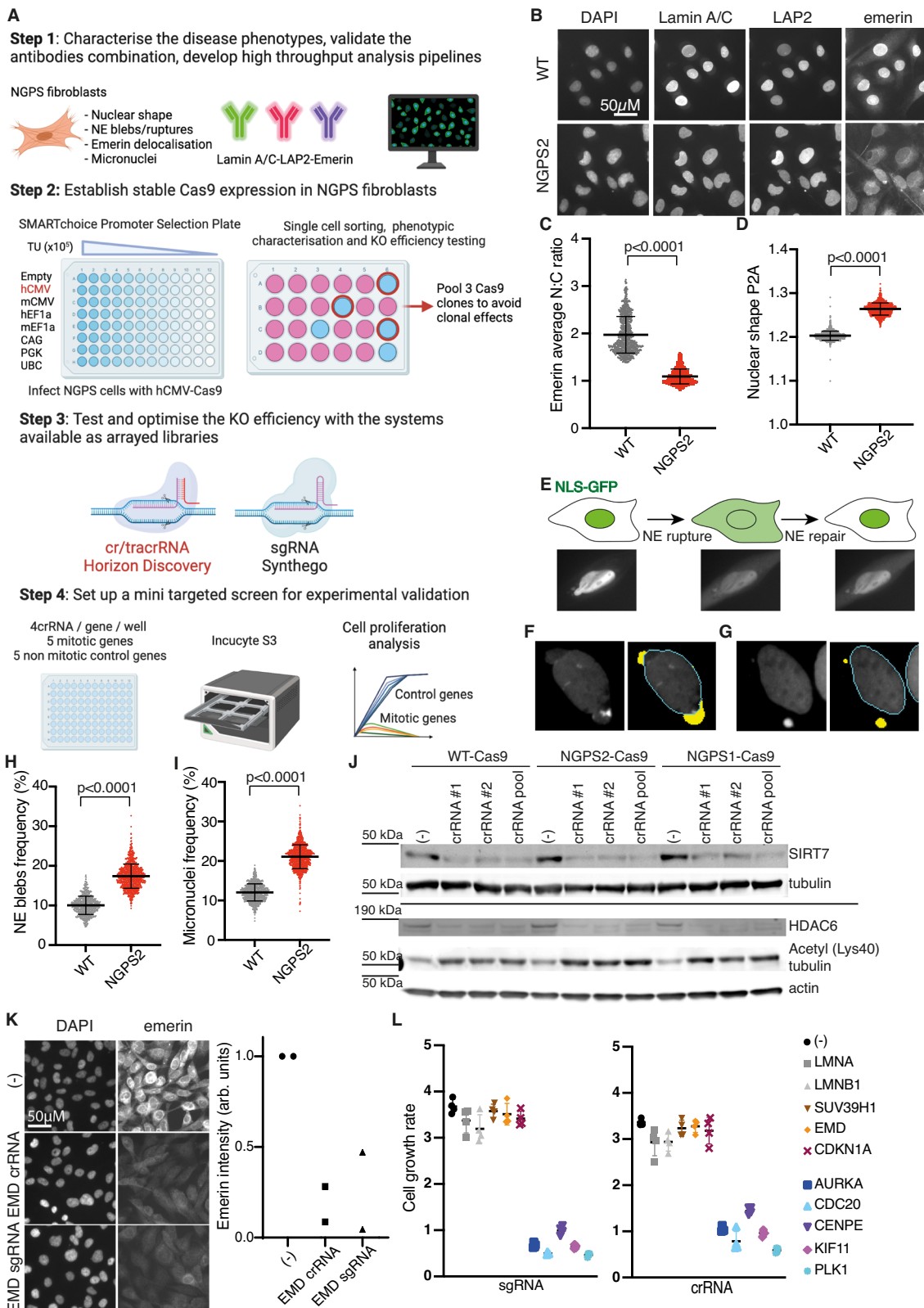

## Depletion of RPS3A, PAFAH1B1, VPS16 and SMU1 suppress the lethality of an NGPS *C. elegans* model

To establish the potential of our screen hits in translating into improvement of NGPS phenotypes in vivo, we used an NGPS *C. elegans* model (hermaphrodites) carrying a *baf-1(G12T)* mutation. We have recently reported the phenotype of these NGPS worms[34]. Interestingly, the NGPS worms have a shorter lifespan and recapitulate the

phenotypes we observed in the patient cells, including loss of emerin from the NE, accumulation of misshapen nuclei, and deregulation of ribosomal genes[34]. This model therefore strongly supports the relevance of the phenotypes and the mechanisms we identified in the patient cells in this study. In accordance with the nucleolar size changes we observed in NGPS cells (Fig. 5E–F), control of nucleolar size appeared to be impaired in *C. elegans* too. Indeed, in hypodermal

**Fig. 2 | Whole genome CRISPR screening set-up in NGPS fibroblasts.**
**A** Schematic detailing the four main steps involved in the CRISPR screen set-up (Created in BioRender. Larrieu, D. (2024) BioRender.com/e32i842). **B** Example immunofluorescence images of the indicated stainings used in the screen and obtained with the high-content microscope. **C** Quantification of the nucleus to cytoplasmic (N:C) emerin intensity ratio in WT and NGPS2 cells. **D** Quantification of the nuclear shape in WT and NGPS2 cells using a perimeter to area (P2A) analysis obtained with HCS Studio. Each data point in (**C**) and (**D**) is the average value measured over 500 cells, in 7 independent experiments; lines indicate the average ± SD and unpaired two-tailed t-tests were used for statistical analysis. **E** Graphical representation and corresponding immunofluorescence images showing the identification of blebs as the origin of NE ruptures using a nuclear localization signal (NLS) reporter tagged with a GFP. **F, G** Representative nuclei imaged with the CX7 microscope, showing the outlines of the nuclei in blue, and the nuclear blebs (**F**) or

micronuclei (**G**) in yellow as detected by the HCS software and quantified (as in **C** and **D**) in (**H** and **I**). Each data point in (**H**) and (**I**) is the average value measured over 500 cells, in 7 independent experiments; lines indicate the average ± SD and unpaired two-tailed t-tests were used for statistical analysis (**J**) Knock down efficiency of individual or pooled crRNAs assessed in the clonal population of Cas9-expressing WT and NGPS cells. crRNAs targeting SIRT7 (top 2 blots) or HDAC6 (bottom 3 blots) were used, either as single sequences (#1, #2) or as a pool of the 2 sequences. **K** Representative immunofluorescence images of emerin intensity in NGPS1-Cas9 cells upon transfection of a pool of 3 crRNAs or sgRNAs and quantified in 2 independent experiments using the HCS image studio software (right panel). **L** Comparison of cell growth inhibition upon transfection of the indicated sgRNA or crRNA in NGPS1-Cas9 cells. The growth rate was measured using an Incucyte S3 live-cell analysis system in 2 independent experiments. The data shows the representative cell growth from 4 images/well, (average ± SD indicated by lines).

nuclei of live NGPS (G12T) *C. elegans* expressing a GFP tagged nucleolar marker, we observed an increased nucleolar area compared to wild-type (WT) worms (Fig. 7A–B). The *baf-1(G12T)* mutant worms are fully viable, but <10% complete development when combined with an endogenously tagged *gfp::lmn-1*/lamin allele (Fig. 7C, Supplementary Fig. 6A). We took advantage of this sensitized background to test for rescue of developmental arrest upon RNAi-mediated knockdown of 32 genes from our screen (Fig. 7C and Supplementary Fig. 6B), for which there was a *C. elegans* homolog. Interestingly in regards to the protein synthesis data we obtained in human cells, depletion of RPS-1 (the *C. elegans* homolog of human RPS3A), a ribosomal protein of the 40S subunit, suppressed the lethality of the *gfp::lmn-1; baf-1(G12T)* animals (Fig. 7B). Three additional genes: *lis-1* (PAFAH1B1)−involved in various dynein and microtubule processes as well as in osteoclast formation[35], *vps-16* (VPS16) −a protein involved in protein trafficking to lysosomal compartments[36], and *smu-1* (SMU1) – involved in mRNA splicing[37,38], led to a similar rescue of worm lethality (Fig. 7C). Depletion of the other genes did not rescue the lethality (Supplementary Fig. 6B), potentially due to some of them−including protein synthesis genes− being essential in vivo.

## Discussion

Our data demonstrates the feasibility and the power of a microscopy based, CRISPR/Cas9 genome-wide screen for suppressing and bypassing progeric phenotypes in patient-derived fibroblasts. While a targeted (320 genes) multiparametric siRNA screen has been carried out previously in a cellular model of HGPS[39], our screening approach has interrogated the entire human protein coding genome (19,200 genes) and used NGPS patient cells. The only patient-derived cells currently available are immortalized fibroblasts, due to the inability of the primary cells to grow in culture (personal communication from Carlos Lopez-Otin). This prevents the possibility to perform functional assays such as replicative lifespan in culture or migration assays which is one caveat of the current study. On the other hand, immortalized cells are much more amenable to large scale screening assays such as the one we performed here, especially when considering the fact that we had to establish Cas9 expressing clones grown from single cells− which would have been very challenging with primary patient fibroblasts. By using both WT fibroblasts from an unaffected individual, as well as NGPS fibroblasts in which we reversed the BAF A12T homozygous mutation using CRISPR/Cas9[14], we were able to identify a reliable combination of four cellular phenotypes that were both specific to the BAF A12T mutation, and quantifiable by high throughput microscopy. This approach allowed us to reduce the number of hits to ~0.2% of the whole genome library, and to identify genes showing the best "rescue" across several phenotypes. In accordance with the principle of synthetic rescue, only 3 of the hits were NE proteins, and more specifically nuclear pore complex proteins: NUP160, SEC13 and AHCTF1 (ELYS). Both SEC13 and ELYS have been previously shown to regulate

nuclear size and nuclear import[40], a function shared by RAN, another hit from our screen. This suggests that modulating nuclear import might have positive effects on NE organization and function in NGPS cells.

In view of the strong skeletal abnormalities of NGPS patients, it is also worth highlighting that our screen identified two genes involved in osteoclast development: *LRRK1* (Leucine-rich repeat serine/threonine-protein kinase 1) and *PAFAH1B1/LIS1* (Platelet-activating factor acetylhydrolase IB subunit alpha), whose depletion improve NGPS cellular phenotypes (Figs. 3 and 5) and in the case of LIS1 also increases viability of NGPS worms (Fig. 7C). LRRK1 has been involved in the regulation of osteoclast activity and bone resorption[41–43]. Accordingly, LRRK1 dysfunction in mice or human is associated with severe osteopetrosis[44]. Similarly, PAFAH1B1 regulates osteoclast formation and bone homeostasis through interacting with the protein Plekhm1, involved in osteoclast secretion. PAFAH1B1 depletion has been shown to strongly reduce osteoclast formation[35]. Therefore, we can hypothesize that the severe osteoporosis and osteolysis observed in NGPS cells might be associated with an increased level or activity of LRRK1 and/or PAFAH1B1. This might contribute to the aberrant NGPS cellular phenotypes that become normalized upon depletion of these genes. As a follow up experiment, it would be interesting to study the effect of the BAF A12T mutation in bone-derived cell lines, and establish the benefit of LRRK1 and PAFAH1B1 depletion to the response of these cells to mechanical stress for example. Once a mouse model for NGPS becomes available, it would be of course also very relevant to test how modulating these genes might improve the phenotypes of the mice in vivo.

Here, we decided to exploit *C. elegans* to perform functional assays, due to the absence of primary NGPS cells (as described above), and because it is a highly appreciated model to study mechanisms of aging. Numerous studies have concluded that interventions that either extend or shorten lifespan in worms have similar effects in other animals, including mammals[45–47]. Secondly, the strong conservation of *BAF* across the animal kingdom both in terms of primary amino acid sequence and predicted secondary structure argues in favor of functional conservation. For instance, it has been shown that phosphorylation of BAF by the kinase VRK1 regulates BAF dynamics similarly in worms, flies, and human cells. In addition and supporting this idea, our recent work[34] shows that the NGPS *C. elegans* model recapitulates phenotypes of the NGPS patient fibroblasts. These include deterioration of nuclear morphology with age, decreased emerin incorporation at the NE, enlarged nucleoli and deregulation of ribosomal genes. These phenotypes are associated with a decreased lifespan and altered resistance to stress.

One of the top functional enrichments in the screen was protein synthesis, with 7 out of the 43 hits being associated with this process. Loss of protein homeostasis is one of the hallmarks of aging[19] and inhibition of translation has been associated with enhanced longevity

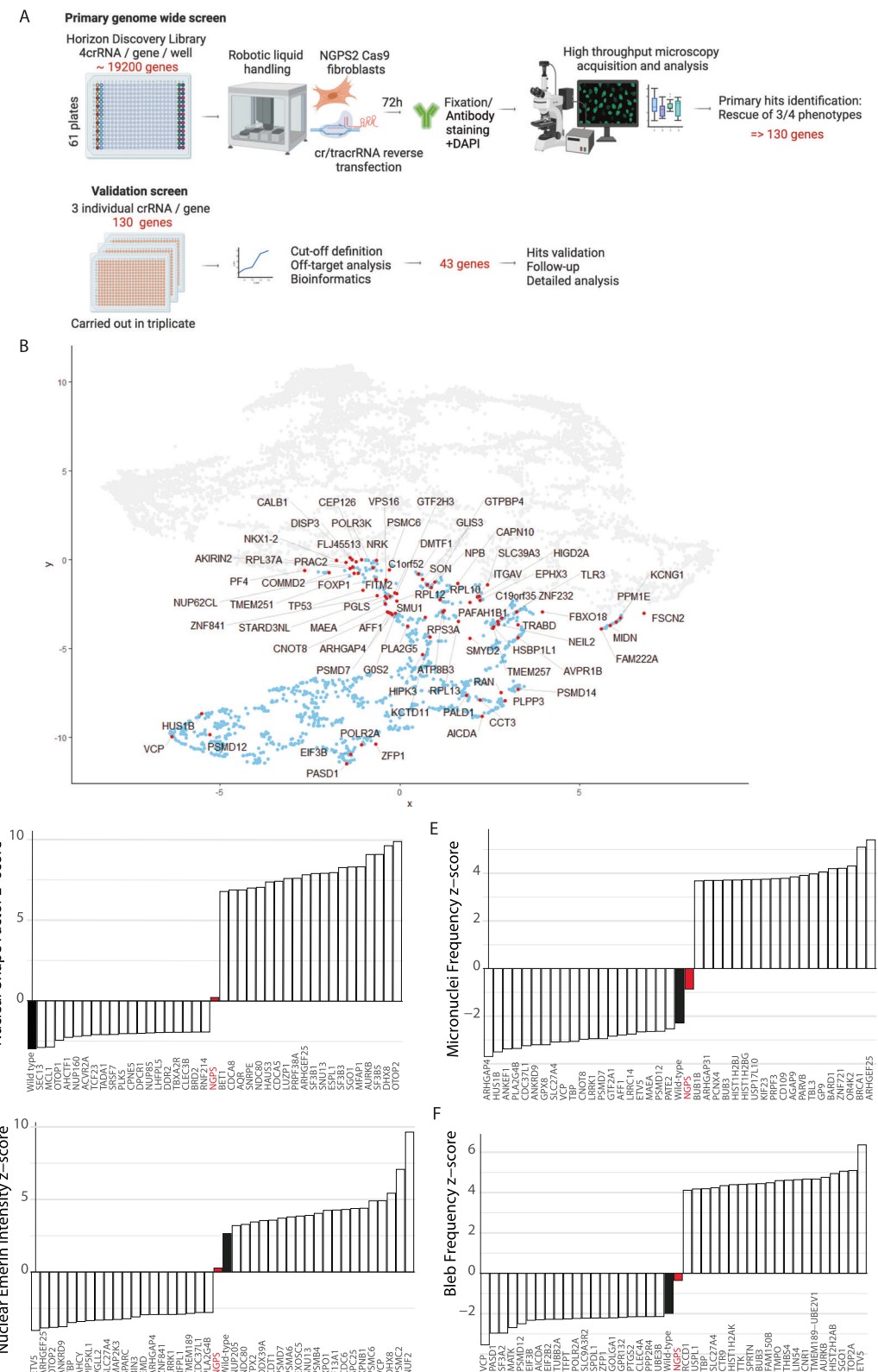

**Fig. 3 | Identification of genes and pathways modulating NGPS phenotypes.**
**A** Schematic showing the experimental outline of the primary and validation screens in NGPS2-Cas9 fibroblasts (Created in BioRender. Larrieu, D. (2024) BioRender.com/e32i842). **B** Uniform Manifold Approximation and Projection (UMAP) of the cluster analysis from the primary screen. Nuclear shape, micronuclei and NE blebs frequency quantified in NGPS2-Cas9 cells (gray dots) for each single knock-out were reduced to two dimensions and mapped alongside the same parameters measured in matching control cells (blue dots). The hit genes are labeled in red. **C–F** S plots depicting the top 20 genes with the highest and lowest Z scores for each individual phenotype quantified from the NGPS genome-wide CRISPR screen alongside the mean value obtained for the NGPS cells (red bars) and the wild type cells (black bars) transfected with a non-targeting crRNA.

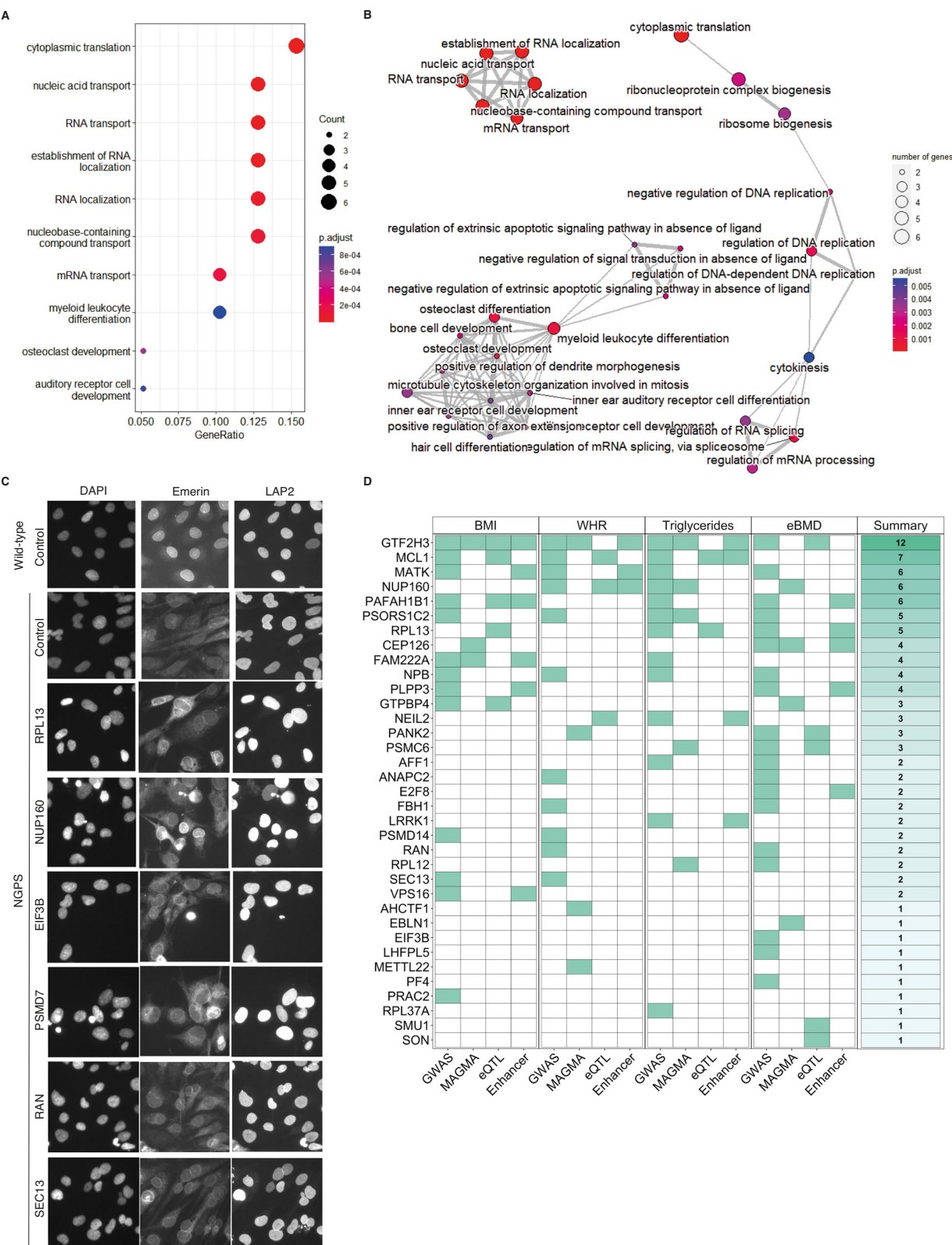

**Fig. 4 | Validation of hits normalizing multiple nuclear envelope phenotypes in NGPS cells. A–B** Biological processes gene set enrichment was carried out using the clusterProfiler R package. *P* values were calculated through clusterProfiling using a hypergeometric distribution which corresponds to a one-sided version of Fisher's exact test. No corrections for multiple comparisons were applied. The 43 validated genes were tested against a full homo sapiens ontology database. **C** Representative immunofluorescence images obtained in the validation screen, showing the effects of some of the hits on the nuclear envelope phenotypes. **D** Heatmap showing the overlap between identified target genes and human

genetic datasets. For all four phenotypic traits (BMI (*n* = 806,834), WHR (*n* = 694,649), triglycerides (*n* = 1,253,275) and eBMD, (*n* = 426,824)), target genes were annotated on the basis of (i) proximity to GWAS signals, (ii) coding-variant gene-level associations to the trait (MAGMA), (iii) colocalization between the GWAS and eQTL data and (iv) the presence of known enhancers within the association peaks. A count of the observed concordant predictors (out of a maximum of 16) is displayed in Summary (right panel). Expanded results can be found in Supplementary Data 8.

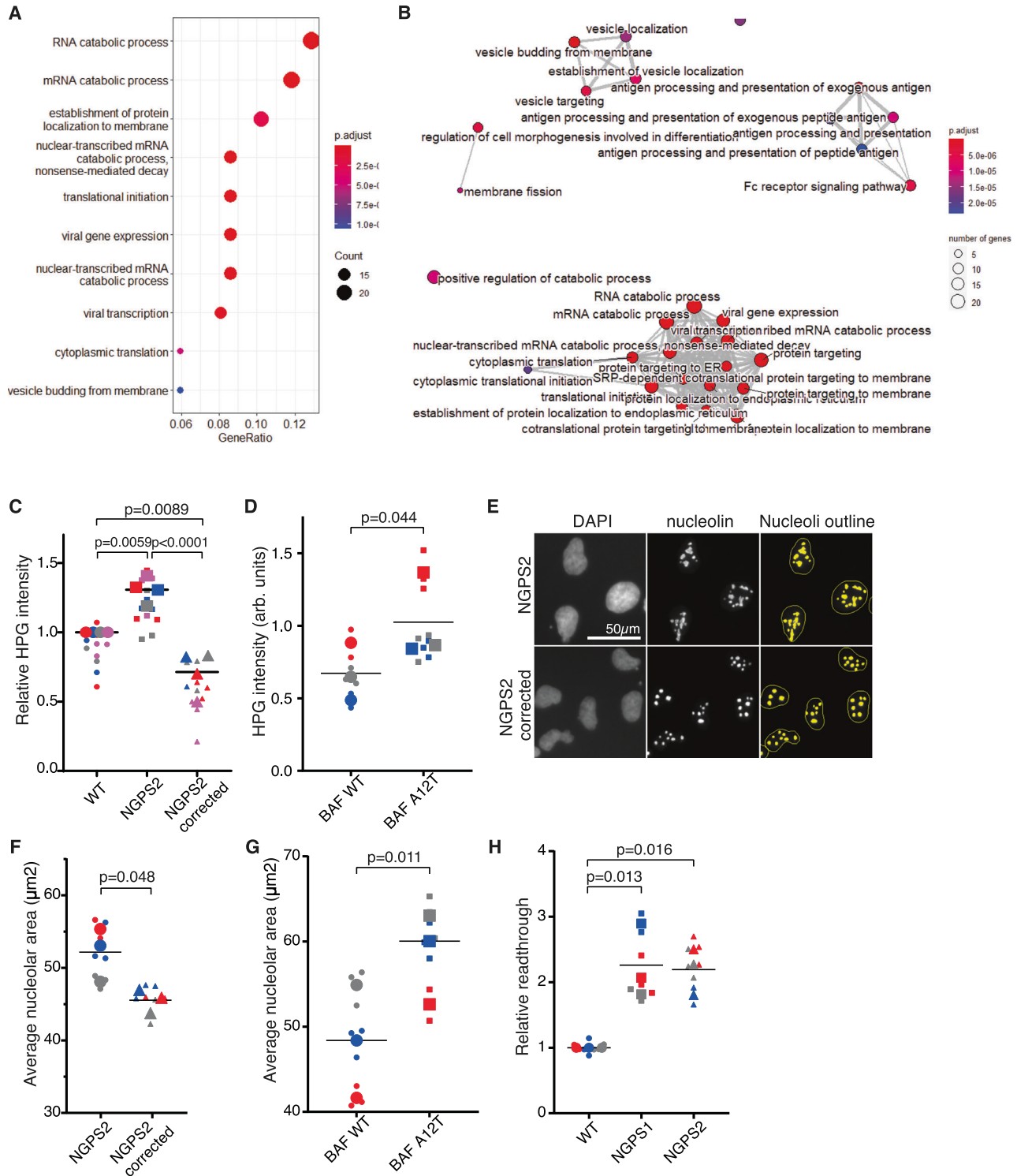

**Fig. 5 | BAF A12T is associated with enhanced protein synthesis and translation errors. A–B** Differential gene expression analysis in wild-type compared to NGPS1 and NGPS2 cell lines reveals enrichment for biological processes associated with RNA processing and translation. *P*-values were adjusted for multiple testing via the False Discovery Rate (FDR; Benjamini-Hochberg) approach. Gene Ontology Analysis was performed on genes with a False Discovery Rate value less than 0.01. **C** Nascent protein synthesis assay using HPG incorporation followed by labeling using a "click" reaction with Alexa fluor 488 (AF 488) in WT, NGPS2 or NGPS2 corrected (**C**) or in WT fibroblasts expressing a BAF WT or BAF A12T construct (**D**). AF 488-HPG intensity was quantified using the high-content microscope in 4 (**C**) or 3 (**D**) independent experiments, each measuring 1000 cells in 3 wells per cell line, graphed as a Superplot with data from different experiments indicated in different colors and larger symbols indicating each experiment's average. AF 488-HPG intensity was normalized to the WT cell line. Mean values were analyzed using nested one-way ANOVA (**C**) or paired two-tailed t-test (**D**), with Tukey's method used for multiple comparisons in (**C**). **E** Immunofluorescence images of nucleolin used to identify and outline the nucleoli (yellow) in the indicated cell lines using the HCS Studio software. **F–G** Average nucleolar area quantified in NGPS2 and NGPS2 corrected cells (**F**) or in WT fibroblasts expressing BAF WT or BAF A12T (**G**). **F** and (**G**) are superplots as in (**C–D**), and statistical comparisons were made using paired two-tailed t-tests. **H** Translation error rate measured as an increased read-through using a dual luciferase assay in 3 independent repeats (Superplot of the data). Mean values were analyzed using one-way ANOVA with Dunnett's multiple comparison testing. Results are derived from the ratio hFluc/hRluc, given in fold induction.

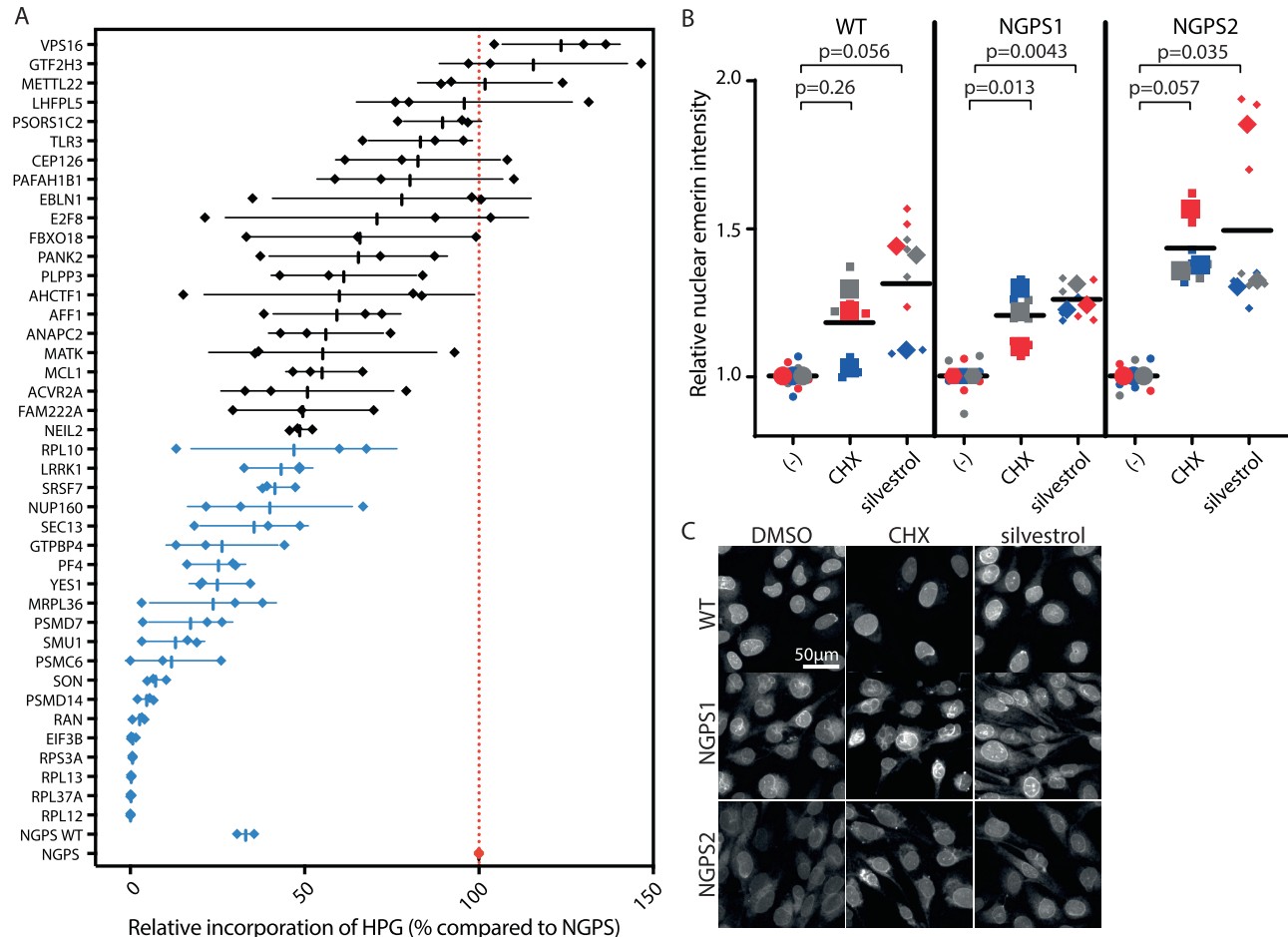

**Fig. 6 | Inhibition of protein synthesis restores nuclear envelope abnormalities in NGPS cells. A** Nascent protein synthesis assay based on fluorescence intensity of HPG AF 488. NGPS2 or NGPS2 corrected cells (NGPS WT) were imaged with the high throughput microscope following siRNA depletion of 41 of the validated screen hits. Depletion of genes highlighted in blue show significant (one-way ANOVA with Dunnett's multiple comparisons test, $p < 0.05$) reduction of HPG incorporation compared to NGPS2 cells transfected with a non-targeting siRNA (red dotted line).

Shown are averages ± SD of 3 independent experiments in 500 cells. **B** Quantification of emerin nuclear intensity in NGPS1 and NGPS2 cell lines compared to WT, upon protein synthesis inhibition using cycloheximide (CHX) or silvestrol. Superplots of the 3 independent experiments are shown, and nested one-way ANOVA was used for statistical comparisons with Dunnett's multiple comparison testing. **C** Representative immunofluorescence images of emerin staining upon protein synthesis inhibition by treatment with CHX or silvestrol.

in several animal models[48–50]. Identifying that inhibition of this pathway was beneficial to NGPS cells and NGPS worms, therefore reinforce the current hypothesis that loss of cellular homeostasis observed in both premature and physiological aging might occur through similar mechanisms. More specifically, we observed that the NGPS associated BAF A12T mutation was associated with a higher rate of nascent protein synthesis, together with differential expression of genes involved in translation regulation (Fig.5A–D). Our recently published paper[34] also shows that the A12T mutation directly affects the DNA binding profile of BAF, leading to differential expression of genes involved in ribosomal structure and translation. Interestingly, almost all ribosomal genes that were found to be differentially expressed in the NGPS worms, associated less frequently with BAF-1(G12T) than with BAF-1 wild-type. This suggests that the BAF A12T mutation may indirectly impact translation by modulating its DNA binding profile, therefore modifying the expression of translation-associated genes. Translation deregulation – potentially occurring through different mechanisms - could therefore be a common process contributing to premature aging, as increased protein synthesis was also observed in primary HGPS cells and was similarly accompanied by increased nucleolar size[32]. In the case of NGPS, we saw that this was associated with a higher rate of errors, as seen by an increased stop codon readthrough. It therefore appears that fibroblasts derived from both HGPS and NGPS

patients go through a phase of enhanced protein synthesis while they are still in a replicative state (low passage number in the case of primary HGPS cells or immortalization in the case of NGPS). As translation is one of the most energy consuming process in the cell, this could accelerate cell exhaustion, contributing to premature entry of the cells into senescence. Once the cells enter into senescence, they then display reduced global protein synthesis. As the NGPS cells are immortalized, we cannot assess the effect of protein synthesis reduction on the replicative lifespan of the cells in culture, and therefore we cannot connect the phenotypic rescue to cellular senescence.

There is currently no mouse model for NGPS but a recent study reported the effect of error-prone protein synthesis in mice[51]. Through engineering a specific ribosomal mutation, the authors induced genome-wide translational errors in mice. This resulted in reduced lifespan and premature aging features resembling the phenotypes observed in NGPS patients including chest and spine deformation as well as loss of fat. This suggests that deregulation of protein synthesis and accumulation of errors might directly contribute to the premature aging phenotypes of NGPS patients. Therefore, identifying non-toxic pharmacological interventions to slow down protein synthesis, thereby potentially limiting error-prone translation might yield phenotypic improvement at the level of the individuals. According to our data, this could be achieved through targeting proteins directly

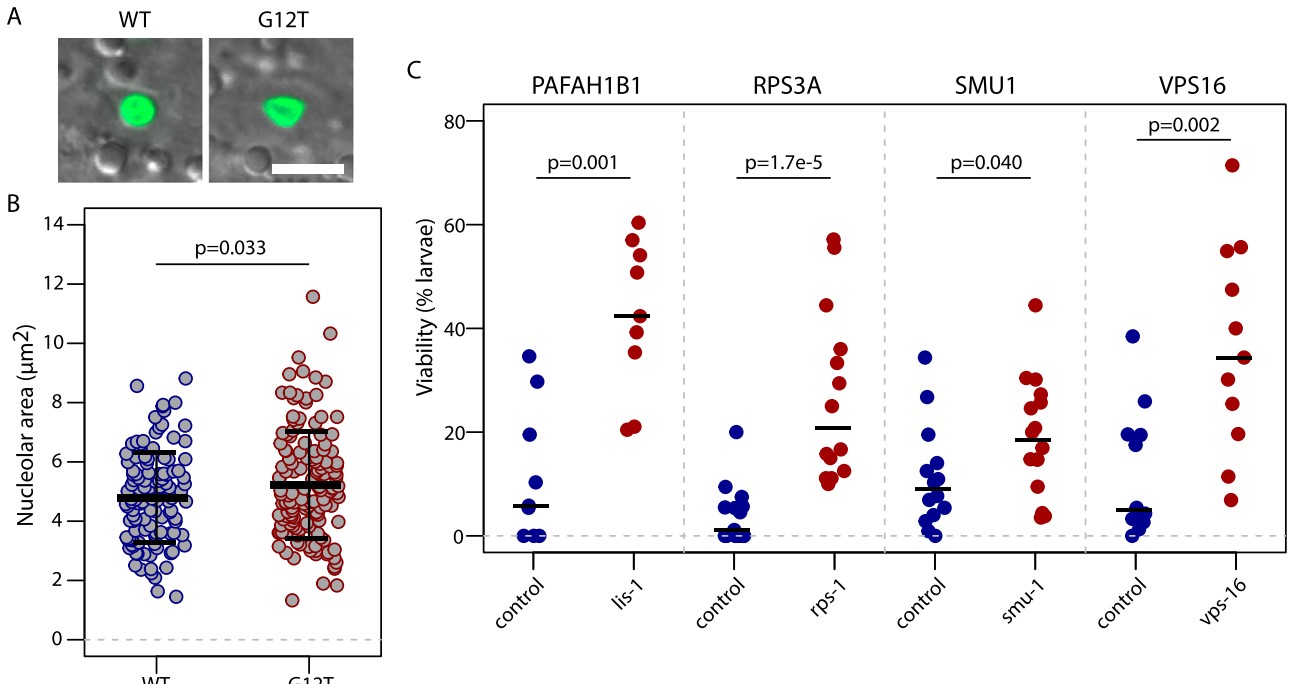

**Fig. 7 | Depletion of PAFAH1B1, RPS3A, SMU1 and VPS16 suppress the larvae lethality of an NGPS hermaphrodite *C. elegans* model. A** Confocal images of hypodermal nuclei of live wild-type (WT; strain COP262) and NGPS (G12T; strain BN1389) 1 day old adult hermaphrodites *C. elegans* expressing FIB-1::GFP. Shown are overlays of GFP (green) and DIC images. Scale bar 5 um. **B** Nucleolar area was measured in 124 nucleoli for the WT worms and 146 nucleoli for the *baf-1(G12T)*, representing > 20 animals per strain over 4 independent experiments. Black lines show the average nucleolar area ± SD. The *p*-value from Welch two sample t-test is indicated. **C** The indicated genes were knocked down by RNAi and tested for suppression of lethality in *gfp::lmn-1, baf-1(G12T)* hermaphrodites (strain BN1336). Each point corresponds to the percentage of eggs developing into larvae from a single plate with 50-100 eggs laid by 1 day old adult hermaphrodites. For PAFAH1B1, 9 plates were analyzed for both the control and PAFAH1B1, over 3 independent experiments. For RPS3A, 15 control plates and 14 RPS3A plates were analyzed over 4 independent experiments. For SMU1, 14 control and 16 SMU-1 plates were analyzed over 5 independent experiments. For VPS16, 12 controls and 11 VPS16 plates were analyzed over 4 independent experiments. Black lines indicate medians. Mann-Whitney test *p*-values were calculated for each set of control and test plates.

involved in protein synthesis (such as RPL proteins) as well as proteins involved in various other cellular functions (Fig. 5G). Indeed, in the cellular context of NGPS, we observed a reduction of nascent protein synthesis upon depletion of genes that don't have any known protein synthesis function. The mechanisms behind this and the link between protein translation and NE integrity remain unknown and will be the basis of future studies, but it does suggest a common mechanism by which depletion of the hits from the screen improve NGPS phenotypes.

Altogether, our screen has shed light on pathways involved in cellular dysfunction caused by the BAF A12T mutation in NGPS patient cells. This work has also identified potential therapeutic targets for this incurable premature aging syndrome, supported by the data we obtained in vivo, showing rescue of lethality in NGPS worms upon knocking down some of the hits from the screen.

## Methods
### Ethics
Our research complies with all relevant ethical regulations. The NGPS-derived patient cells were obtained from Prof. Lopez-Otin with informed consent from the patients and ethical oversight.

**Cell culture.** Wild type (WT, derived from an age-matched healthy individual (AG10803, Coriell repositories), NGPS5796 (NGPS 1) and NGPS5787 (NGPS 2) −both males−immortalized fibroblasts were a kind gift of Prof. Carlos López-Otín (University of Oviedo, Spain). All three cell lines were immortalized with SV40LT and TERT. Cells were grown at 37 °C in a 5% $CO_2$ incubator in Dulbecco's modified Eagle's medium containing 4.5 g/L glucose (Sigma) supplemented with 2 mM L-glutamine (Sigma), 10% fetal bovine serum, 100 units/mL penicillin and 0.1 g/L streptomycin (complete medium). WT cells expressing

Flag-BAF or Flag-A12T BAF were described before[14] and maintained in complete medium containing 100 µg/mL hygromycin B (Toku-E, #H007, made up to 100 mg/mL in PBS). Cells were passaged twice a week and used within 12 passages. Cells were free of mycoplasma as assessed using strips (InvivoGen #rep-mys-20). For the phenotypic comparison with HGPS (Fig. 1 and S1), WT cells were GM05565 fibroblasts and HGPS cells were AG11513 cells, both obtained from Coriell Institute for Medical Research (Camden, New Jersey, U.S.A.).

**Antibodies and reagents.** Supplementary Table 1 lists antibodies and reagents used in Western blotting (WB) and immunofluorescence (IF) applications.

**Immunofluorescence.** Cells were seeded on thickness 1 ½ round coverslips (Thermo Fisher Scientific) in 12-well plates for full-field imaging, thickness 1 ½ square (18 mm × 18 mm) coverslips (Zeiss) in 6-well plates for super-resolution imaging or in a 96-well view-plate (PerkinElmer #6005182) for high-content imaging and grown to a confluency level consistent across the cell lines to be compared. All subsequent steps were carried out at room temperature. Cells were fixed with 4% paraformaldehyde in PBS (Boster, AR1068) for 20 minutes, followed by permeabilization in 0.2% Triton X-100 in PBS for 12 minutes. Unspecific antibody binding was blocked by incubating in 5% BSA (Sigma #A7906), 0.1% Tween-20 in PBS (IF blocking buffer) for 30 minutes. Primary antibodies were diluted in IF blocking buffer (see Supplementary Table 1) and incubated for 1–2 hours. After 3 washes with PBS cells were incubated with secondary antibodies and DAPI diluted (see Supplementary Table 1) in IF blocking buffer. After 3 washes with PBS coverslips were mounted using Prolong Gold (P10144) and left to set at room temperature overnight; cells stained in

96-well plates were overlaid with 100 µL/well fresh PBS and stored at 4 °C until imaging. Wide-field immunofluorescence imaging was on an upright Axioimager Z2 (Zeiss) using a Hamamatsu Flash 4 sCMOS camera and Zeiss plan-apox lens, 63 × 1.4 N.A. oil immersion objective and ZenBlue 2012 image acquisition software. Super-resolution imaging was on an Elyra PS1 structured illumination microscope (Zeiss) using a ZenBlack SR edition image acquisition software (Zeiss), PCO edge 4.2 sCMOS camera and 63 × 1.4 N.A. oil immersion objective. High-content imaging was on a CellInsight CX7 microscope (Thermo Fisher Scientific) using a 20 × 0.35 N.A. objective. HCS Studio software 2021 (Thermo Fisher Scientific) was used for quantitative image analysis. For quantitation of H3K9me3 or HP1γ nuclear intensities the DAPI images were used to define the nuclear contour, and a fixed intensity threshold was set to define the stained heterochromatin domains.

**Western blotting.** Confluent monolayers of cells in 6-well plates were washed with PBS and scraped in 70 µL/well SDS lysis buffer (4% SDS, 20% glycerol, and 120 mM Tris-HCl (pH 6.8)). Lysates were incubated for 5 min at 95 °C. The DNA was sheared by syringing 10 times through a 25-gauge needle. Absorbance at 280 nm was measured (NanoDrop, Thermo Fisher Scientific) to determine protein concentration. Samples were prepared in NuPAGE sample buffer (Thermo Fisher Scientific #NP0007) and DTT (100 mM) and heated at 95 °C for 10 min. Proteins were loaded on NuPAGE 4-12% Bis-Tris gels (Thermo Fisher Scientific), separated in NuPAGE MES SDS running buffer (Thermo Fisher Scientific #NP0002) and transferred to 0.2 µm pore size nitrocellulose membranes (Amersham, #10600004) for immunoblotting. Blotted proteins were reversibly stained with Ponceau S solution (Thermo Fisher Scientific, # A40000279) to allow cutting strips for individual antibody incubations. Membrane strips were first blocked for 30 min in 5% milk in TBST buffer (20 mM Tris, 150 mM NaCl, 0.1% Tween-20) before incubation with primary antibodies diluted in TBST buffer (see Supplementary Table 1) for 1 h at room temperature. After 3 washes with TBST buffer membrane strips were incubated for 1 h with IRDye-conjugated secondary antibodies (LI-COR, see Supplementary Table 1). After 3 more washes with TBST buffer membrane fluorescence was scanned on an Odyssey CLx imaging system (LI-COR).

**Transmission electron microscopy.** Cell monolayers were fixed by the addition 2.5% glutaraldehyde / 2% paraformaldehyde in 0.1 M Na cacodylate buffer, pH 7.2 at 37 °C. The monolayer was then scraped from the tissue culture plastic, pelleted in a benchtop microfuge and allowed to cool to room temperature.

The cell pellet was washed with 0.1 M Na cacodylate buffer, pH 7.2; post-fixed in 1% osmium tetroxide in 0.1 M Na cacodylate buffer, pH 7.2, for 1 hour, and washed with 0.05 M Na maleate buffer, pH 5.2. Next, the cells were stained en bloc with 0.5% uranyl acetate in 0.05 M Na maleate buffer, pH 5.2, for 1 h at 4 °C; washed with 0.05 M Na maleate buffer, pH 5.2; dehydrated in a graded series of ethanol and exchanged into 1,2-epoxy propane. The cell pellet was then infiltrated with 50:50 epoxy propane: Agar 100 resin overnight before exchange into Agar 100 resin. Finally the cell pellets were embedded in Agar 100 resin in BEEM capsules (Agar Scientific, Stansted, UK) overnight at 60 °C. Ultrathin sections (60 nm) were cut using a diamond knife mounted on a Leica Ultracut UC7 ultramicrotome (Leica, Milton Keynes, UK), collected on formvar-coated copper EM grids and stained with uranyl acetate and Reynolds lead citrate. The sections were observed in an FEI Tecnai G2 Spirit BioTWIN transmission electron microscope (Eindhoven, The Netherlands) at an operating voltage of 80 kV. Images were captured using a Gatan US1000 CCD camera.

**Generation of Cas9-expressing cells.** Stable Cas9 expression was engineered in the WT and in both NGPS cell lines as follows. We first assessed protein expression efficiency from seven constitutive promoters (hCMV, mCMV, hEF1a, mEF1a, CAG, PGK and UBC) using a

SMARTchoice promoter selection plate (Horizon Discovery, #SP-001000-01). The hCMV promoter showed the highest expression and was chosen to drive Cas9 expression. WT and NGPS cells were transduced with purified lentiviral particles, containing a vector encoding the S. pyogenes Cas9 nuclease under the control of a hCMV promotor according to the manufacturer's protocol (Horizon Discovery, #VCAS10124). Cells stably expressing Cas9 were selected using 5 µg/mL blasticidin (Merck/Sigma Aldrich), and several clones were isolated from the polyclonal population by single cell sorting. Individual clones were assessed for Cas9 expression, cellular morphology, effect on cell health markers such as DNA damage (gH2AX level) and proliferation, as well as Cas9 cutting efficiency. The latter was assessed based on resistance to 6-thioguanine upon HPRT knock out as described before[31] as well as by knock down efficiency using immunofluorescent staining or Western blotting (see CRISPR mini-libraries and Supplementary Table 2). Based on these, to avoid any clonal effect that might arise from a single clone, three Cas9-expressing clones were selected for each cell line and pooled for further CRISPR experiments (WT-Cas9 or NGPS-Cas9) including the mini-libraries and genome-wide screens.

**CRISPR mini-libraries.** Two CRISPR mini libraries were obtained; one contained pools of 3 sgRNAs targeting 10 genes (Synthego, Redwood City, CA, USA) (Supplementary Table 3), the other contained pools of 4 crRNAs targeting 20 genes (Horizon Discovery, Waterbeach, UK) (Supplementary Data 1). For immunofluorescence or cell proliferation assays, WT-Cas9 or NGPS-Cas9 cells were reverse transfected with the CRISPR reagents in a 96-well view-plate (Perkin Elmer #6005182). Per well 20 µL transfection mix was first prepared in a V-bottom 96-well plate (Greiner #651161) as follows. For crRNA:tracrRNA transfection, 2.5 µL 1 µM crRNA and 2.5 µL 1 µM trRNA were added to 5 µL Optimem; for sgRNA transfection 2.5 µL 1 µM sgRNA was added to 7.5 µL Optimem. To each well containing 10 µL diluted crRNA:tracrRNA or sgRNA was then added 10 µL of a Dharmafect-1 dilution (0.05 µL Dharmafect-1 in 9.95 µL Optimem). The solutions were mixed by pipetting up-and-down and incubated for 20 minutes at room temperature before arraying in a 96-well view-plate (Perkin Elmer #6005182). Meanwhile, cell suspensions were counted using a Countess cell counter (Thermo Fisher Scientific) and dilutions prepared such that to each well were added 3300 cells for WT-Cas9 and NGPS2-Cas9 or 5000 cells for NGPS1 cells in a 80 µL volume in antibiotic-free growth medium. Plates were shaken by hand and placed in a 37 °C incubator for 65-72 hrs. For cell survival analysis plates were put in an Incucyte live-cell imaging incubator 24 hrs post-transfection (see Cell proliferation). For Western blotting analysis of sgRNA-transfected cells, reverse transfections were carried out in 12-well format. Per well 3 µL 10 µM sgRNA was diluted in 50 µL Optimem, incubated for 5 minutes before adding diluted Dharmafect-1 (0.6 µL Dharmafect-1 in 50 µL Optimem). Transfection mixes were overlaid with 900 µL antibiotic-free growth medium containing 120,000 NGPS1-Cas9 cells. Plates were shaken and put at 37 °C for 72 hrs before cell lysis.

**Cell proliferation (Incucyte).** WT or NGPS cells were seeded onto 24-well plates (Falcon, # 353047), placed in a live-cell imaging incubator (Incucyte S3, Essen, Germany) and imaged with a 10x phase objective every 4 hours for 96 hours. Incucyte S3 analysis software was used to measure and quantify cell density over time. For assaying the mitotic gene knock-outs in the CRISPR mini-libraries WT-Cas9 or NGPS-Cas9 cells were reverse transfected in 96-well view-plates (Perkin Elmer #6005182) and placed in the Incucyte imaging incubator 24 hours post-transfection.

**CRISPR library.** A genome-wide CRISPR library was obtained from Horizon Discovery (GP-004650-E2-01, GP-004675-O2-01 and GP-006500-O2-01). The library was arrayed in 61 384-well plates and contained 4 unique crRNAs per gene per well, targeting a total of 19,127

human genes (Supplementary Data 2). Some crRNA pools were present in duplicate or triplicate, providing internal controls for the screen. The 0.1 nmole crRNA pools were resuspended in 20 μL 10 mM Tris buffer (Horizon Discovery #B-006000-100) to yield a 5 μM master plate. This master plate was further aliquoted into mother plates. To one mother plate containing 2.4 μL/well crRNA was added 21.6 μL 0.56 μM tracrRNA (Horizon); the resulting 24 μL of 0.5 μM crRNA:tracrRNA solution was then divided over transfection-ready daughter plates, each containing 6 μL of 0.5 μM crRNA:tracrRNA per well. All liquid-handling steps were carried out using a CyBio Felix liquid handling robot (Analytik Jena, Jena, Germany). Plates were spun for 2 minutes at $300 \times g$, sealed using clear polypropylene seals (Starlabs E2796-0793) and stored at -30 °C.

**Genome-wide CRISPR screen.** The genome-wide CRISPR screen was carried out with the NGPS2-Cas9 pool of 3 clones. Library plates containing transfection-ready crRNA:tracrRNA complexes were thawed and spun at $300 \times g$ for 3 minutes. Emerin and non-targeting crRNA:tracrRNA complexes were added to columns 23 and 24 (see plate layout, Figure. S3A) as positive controls for the transfection efficiency and negative controls respectively. For reverse transfection, Dharmafect-1 (Horizon Discovery) was diluted in Optimem (Thermo Fisher) (120 μL in 23.88 mL Optimem) and incubated for 5 minutes at room temperature before distributing in a 384-well plate. A liquid handling robot (CyBio Felix, Analytik Jena, Germany) was used to add the Dharmafect 1 solution to the crRNA:tracrRNA complexes and to mix the 2 solutions. After 20 minutes of incubation at room temperature, the transfection mixes were distributed in 384-well imaging plates (Greiner #781182) and overlaid with 1200 NGPS2-Cas9 cells/well in 30 μL of complete DMEM medium. WT cells and NGPS2 WT cells (from the 2 clones in which the BAF A12T mutation was reversed using CRISPR Cas9[14] −see Figure. S3A) were added in columns 1 and 2 of each plate (that do not contain transfection complexes). Plates were then kept in a 37 °C incubator with 5% $CO_2$ for 72 hours. All subsequent steps were carried out at room temperature. Cells were fixed using a Wellmate robot (Thermo Fisher) by adding 30 μL/well of 4% PFA for 30 minutes. Fixative was removed through plate inversion, cells were washed once with 40 μL/well PBS and permeabilized with 40 μL/well of 0.2% Triton X-100 in PBS for 12 minutes. Unspecific antibody binding was blocked by incubation for 30 minutes with IF blocking buffer (see Immunofluorescence). Next, cells were incubated for 1 hour with a mixture of primary antibodies (20 μL/well) diluted 1:1000 in IF blocking buffer: rabbit anti-emerin, mouse IgG$_{2b}$ anti-lamin A/C and mouse IgG$_1$ anti-LAP2 (details in Supplementary Table 1). Following a wash with PBS (40 μL/well), cells were incubated for 1 hour with a mixture of secondary antibodies (Alexa Fluor 488 anti-mouse IgG$_{2b}$, Alexa Fluor 568 anti-mouse IgG$_1$ and Alexa Fluor 647 anti-rabbit − details in Supplementary Table 1), as well as DAPI (Supplementary Table 1), in IF blocking buffer (20 μL/well). After a final wash with 40 μL/well PBS the cells were overlaid with 40 μL/well fresh PBS, plates were sealed with a black seal (Perkin Elmer) and loaded onto a OrbitorRS plate handling robot (Thermo Fisher) for loading into and imaging using a CellInsight CX7 high-content microscope (Thermo Fisher). 4-channel images were acquired with fixed exposure times using a $20 \times 0.35$ N.A. air objective for 9 fields/well or until 500 nuclei ("objects") were detected in the DAPI acquisition channel. The HCS studio colocalisation bio-application was used to calculate a nuclear shape parameter for the DAPI-defined objects and the emerin intensity in both the nucleus, defined by the DAPI mask, and in a cytoplasmic ring defined by expansion of the DAPI mask. For micronuclei analysis, the LAP2 images were run through the HCS studio micronuclei bio-application. This analysis module was further modified to detect nuclear blebs, again using the LAP2 images[52]. All primary screen results are collated in Supplementary Data 3.

**Primary screen hit identification.** Two multivariate analyzes were carried out in R using custom scripts which are described here: Raw genome-wide CRISPR screen data were read into R in a plate-by-plate basis. Control wells were annotated before normalization of emerin nuclear intensity, emerin ratio and nuclear shape factor P2A values to the median of the corresponding value of the NGPS2 negative control on each plate. Z scores were calculated (calculated by subtracting the mean of each phenotype from each value and dividing by the standard deviation of each phenotype) and selected values plotted in S plots (Supplementary Figs. x, y). For both multivariate analyzes, data dimension reduction was achieved using UMAP[53] and clustering using k means, with optimal cluster number determined by manual determination of differential grouping of WT control samples to all other samples. A four-parameter (average nuclear emerin intensity, micronuclei frequency, nuclear shape factor P2A, and nuclear bleb frequency) analysis, including a scaling step prior to dimension reduction (subtraction of the values of each parameter by the mean of that parameter followed by division of the values of each parameter by the standard deviation of each parameter), yielded 51 hits. A three-parameter analysis, excluding emerin nuclear intensity as few gene knockouts rescued the phenotype, yielded 68 hits. Of the combined 119 hits we excluded genes whose knockout negatively affected cell survival or proliferation (nuclear count < 200), and we added genes not yet on the list that appeared in the top 15 of three individual phenotypes (nuclear shape factor, micronuclei or nuclear envelope bleb frequency), bringing the total number of hits taken forward for a deconvoluted validation screen to 130 (Supplementary Data 4). The scripts used in this study have been deposited on Zenodo and can be found here: https://doi.org/10.5281/zenodo.13927778.

**Validation screen.** For each of the 130 primary screen hits, 3 crRNA sequences were arrayed randomly in the central wells of two 384-well plates as a custom library (Horizon Discovery) (Supplementary Data 5). Control crRNA:tracrRNA complexes were added to the plates as detailed in Figure. S3B. Transfection, fixing, staining and imaging were all carried out as for the primary screen and repeated 3 times. Aggregate z-scores were calculated for the 3 sequences/genes for each phenotype (Supplementary Data 6). Genes whose z-score was below (for micronuclei frequency, NE bleb frequency and nuclear shape parameter P2A) or above (for emerin nucleus:cytoplasm intensity ratio) the z-score for the NT control in the 3 repeat experiments were considered validated. The 43 validated genes are in the STRING diagram in Fig. S3C.

**Nascent protein synthesis assay.** Nascent protein synthesis was assayed using incorporation of the methionine analog L-homopropargylglycine (HPG). Cells were grown in methionine-free medium (Thermo Fisher Gibco #21013-024) for 1 hour before incubation with 1 μM click-iT HPG (Thermo Fisher #C10428) in methionine-free medium for 1 hour at 37 °C. All subsequent steps were carried out at room temperature. Cells were fixed with 4% paraformaldehyde in PBS for 20 minutes, followed by permeabilization with 0.2% Triton X-100 in PBS for 12 minutes. Next, a click reaction was allowed to proceed for 1 hour by incubation with 4 mM CuSO4, Alexa Fluor 488 azide (1:1000 dilution) and 10 mM sodium ascorbate in the dark. Click reagents were washed off and nuclei were stained with 0.2 μg/mL DAPI before imaging on the high-content microscope.

**siRNA transfection.** A custom siRNA mini-library containing On TargetPLUS siRNA smartpools (Horizon Discovery) targeting 41 of the 43 validated hits as well as some internal control genes (EMD, LMNA and TMPO (encoding LAP2)) arrayed in a 96-well plate was obtained (Supplementary Data 7, see Figure. S4F for the plate layout). The 0.1 nmole siRNA/well was diluted to 2 μM with 10 mM Tris buffer (Horizon Discovery #B-006000-100). Transfection mixes were prepared by

adding diluted Lipofectamine RNAiMAX (Thermo Fisher Scientific) (1 µL in 33 µL Optimem) to diluted siRNA (5 µL 2 µM siRNA added to 28 µL Optimem) in a V-bottomed 96-well plate (Greiner #651161), mixing and incubating for 10 minutes at room temperature. 20 µL of the transfection mix was then transferred to a 96-well viewplate well (Perkin Elmer #6005182) and overlaid with 80 µL cell suspension (NGPS2 cells, 3000 cells/well). siRNA transfection was allowed to proceed for 72 hrs before cells were assayed for nascent protein synthesis as detailed above.

**Protein translation inhibition studies.** WT or NGPS cells were seeded in 96-well imaging plates (PerkinElmer #6005182) and treated for 72 h with 0.125 µg/mL cycloheximide (CHX, Sigma # C4859) or 2.5 nM silvestrol (Biovision #B2417-100) before staining and imaging as in the screens.

**Protein mistranslation assay.** WT or NGPS cells were seeded in 12-well plates. The next day cells were transfected with plasmid pRM hRluc-hFluc D357X, where Asp$^{357}$ (GAC codon) was replaced by a UGA nonsense codon in the firefly luciferase (Fluc) transcript[33], using TransIT-2020 (Mirus #MIR5404) according to the manufacturer's protocol. 30 h post-transfection cells were lysed in 200 µL/well passive lysis buffer (Promega #E1941) for 15 minutes at r.t. under gently rocking. Lysates from each 12-well were collected in 3 wells of a black 96-well plate (PerkinElmer #6005182) on ice. The activities of firefly and sea pansy (Renilla) luciferases were measured sequentially using reagents of the dual-luciferase reporter (DLR) system (Promega #E1910) with a ClarioStar plate reader (BMG Labtech Ltd., Aylesbury, U.K.). In brief, firefly luciferase activity was assayed through addition of 60 µL/well Luciferase Assay Reagent II. Next, firefly luciferase activity was quenched and Renilla activity measured by addition of 60 µL/well Renilla substrate in Stop&Glo buffer. As a positive control for translation errors cells were treated for 24 h with 0.5 mg/mL G418 (Gibco # 10131035) prior to assaying.

**RNA sequencing and analysis.** Total RNA was extracted from wild type, NGPS1 and NGPS2 cells using the RNA extraction kit from Zymo Research. The RNA was analyzed by Cambridge Genomics Services with Illumina TruSeq Stranded mRNA library prep kit. Input for libraries was 1 µg for each sample. After the libraries were prepared, they were pooled in equal quantities and Quality Control was run including Bioanalyzer (average size 283 bp) and qPCR (concentration 43 nM). The RNAs were then sequenced using an Illumina NextSeq 500 2 × 75 cycles MID output kit. 2 pM of pooled libraries were loaded onto the sequencer and 1% Phix was added. FastQC (v0.11.4) was used to assess sequence quality and nucleotide content of paired-end fastq files and enabled trimming of low quality bases from the 3' end, using TrimGalore (v0.5.0). Trimmed reads were then aligned with STAR (v2.7.9) and HISAT2 (v2.1.0), for genome and transcript level analyzes respectively, using Ensembl's Homo sapiens GRCh38 (release 109) reference files. Uniquely aligned genomic reads, with a mapping score of 10 or greater, were quantified using HTSeq (v0.6.1), while StringTie (v1.3.4) was used to assemble and estimate transcript expression levels of HISAT2 alignments. Read counts were then read into R statistical software (v 3.6.1) and used as input to the edgeR package (v3.26.5) for pairwise comparisons, to identify differentially expressed genes and transcripts between Wild type cells and NGPS1/NGPS2 cells. A '5 CPM (Counts Per Million) in at least half of the samples' filter was applied to the counts, which were then normalized before differential tests were performed. The CQN Bioconductor package (v1.30.0) was also used to correct for any Guanine-Cytosine or gene-length biases, prior to pairwise comparisons. Finally, p-values were adjusted for multiple testing via the False Discovery Rate (FDR; Benjamini-Hochberg) approach. Gene Ontology Analysis was performed on genes with a False Discovery Rate value less than 0.01.

**Statistics.** Statistical analysis was done using Graphpad Prism v9. The statistical test used is indicated in the figure legends. Asterisks in the figures correspond to p-values as follows: $*0.01 \leq p < 0.05$, $**0.001 \leq p < 0.01$, $***0.0001 \leq p < 0.001$, $****p < 0.0001$. ns indicates a non-significant statistical comparison.

**Genome wide association studies (GWAS).** Target genes from the whole genome screen were integrated with common variant genome-wide association studies (GWAS) and associated functional annotations, pertaining to coding variants, expressions quantitative trait loci (eQTL) datasets and proximal enhancers.

For the common variant GWAS, we used data on body mass index (BMI, $n = 806,834$) and waist-hip ratio (WHR) adjusted for BMI ($n = 694,649$) from the GIANT study[54], the GLGC triglycerides study[55] ($n = 1,253,275$) and the GEFOS estimated bone mineral density (eBMD, $n = 426,824$) study[56] and only variants with a minor allele frequency > 0.1%. For each of the target genes and each phenotypic trait, genes were annotated based on proximity to genome-wide significant signals ($p < 5 \times 10^{-8}$), in 1 Mb windows; 500 kb up- or downstream of the genes start or end site. As most GWAS signals are intronic or intergenic, we overlayed these associations with other functional datasets to understand whether the associated variants can be causally linked to changes in the identified genes' regulation or expression. First, we calculated genomic windows of high linkage disequilibrium (LD; $R^2 > 0.8$) for each given signal and mapped these to the locations of known enhancers for the target genes, using the activity-by-contact (ABC) enhancer maps[57], to indicate whether the genomic variants associating with the traits of interest directly changed the sequence of enhancers for the genes in question. We then performed colocalisation analyzes between the four GWAS and expression quantitative trait loci (eQTL) data for genomic variants reaching at least a suggestive level of significance in the GWAS ($p < 5 \times 10^{-5}$), using the SMR and HEIDI tests (v1.02[58],) and blood gene expression level data from the eQTLGen study[59]. In doing so we essentially matched the pattern of association observed between the identified genomic variants and the GWAS outcome to the association towards the measured transcript level changes for the identified genes, to approximate the direct effect of the associated variants on the target gene expression. We considered gene expression of a gene to be influenced by the same genomic variation as that seen in the GWAS, if the FDR-corrected p-value for the SMR-test was $p < 0.05$ and the p-value for the HEIDI test was > 0.1%. Both of these sets of results allow us to draw methodological hypotheses about how genetic variants at the identified loci, can cause changes in the observed phenotypic traits, by directly affecting the genes regulation and expression patterns. Finally, we wanted to identify potential coding variants that might also associate with variation in the four phenotypes, indicating more direct protein-level consequences to the identified genes. To do this, we collapsed common coding variants within each of the identified genes and calculated gene-level associations towards each of the four traits, using a gene-level MAGMA analysis (v1.09) and SMR-HEIDI (v0.6886)[60]. Genes exhibiting an FDR-corrected MAGMA p-value < 0.05 were considered significant.

***C. elegans* maintenance.** BN808 baf-1(bq19[G12T]) III was generated by backcrossing baf-1(G12T) mutants (six times with the N2 wild type strain. The baf-1(bq19[G12T]) allele contains two nucleotide substitutions at position 34-35 relative to the start codon (GG®AC) introduced by CRISPR/Cas9 genome engineering. BN1336 yc32[gfp::lmn-1] I; baf-1(bq19[G12T]) III was generated by crossing BN808 and UD484[61] and balanced with hT2 [bli-4(e937) let-?(q782) qIs48] (I;III)). BN1389 knuSi221[fib-1p::fib-1::gfp + unc-119(+)] II; baf-1(bq19[G12T]) III was obtained by crossing BN808 and COP262[62]. These hermaphrodite strains were maintained at 16°C on solid Nematode Growth Medium (NGM) plates seeded with *Escherichia coli* OP50 bacteria[63].

***C. elegans* nucleolar size measurement**. Nucleolar area was quantified in the COP262 and BN1389 strains using the FIB-1::GFP reporter. Larvae were raised at 16 °C and synchronized by picking L4 hermaphrodites and further incubated for 24 h at 25 °C. Young adults were anaesthetized in a drop of 10 mM levamisole on top of soft agar pads as described[64]. Stacks of confocal images were acquired using a Nikon Eclipse Ti microscope equipped with Plan Fluor 40×/1.3 and Plan Apo VC 60×/1.4 objectives and an A1R scanner using a pinhole of 1.2 airy units. Nucleolar areas were measured in 124 nucleoli for the WT worms and 146 nucleoli for the *baf-1(G12T)*, representing > 20 animals per strain over 4 independent experiments, after creating binary masks with Fiji software from the original images[65].

***C. elegans* RNAi**. For RNA interference, were fed with E. coli that express double-stranded RNA (dsRNA). Bacterial clones were obtained from a genome wide RNAi library[66] except clones corresponding to *mel-28, npp-6, npp-20, rpn-9* and *rpl-13* that were obtained from alternative sources[67–69]. RNAi plasmids for *csk-1, daf-4, gft-2H3, src-1, ced-9* and *mrpl-36* were constructed as described[70] using the PCR primers listed in Supplementary Table 2. PCR fragments were inserted into plasmid pL4440 npp-15[68] after digestion with XhoI (NEB R0146S) and SpeI (NEB R3133S), ligation and subsequent transformation into E. coli DH5α. Plasmids from ampicillin resistant colonies were confirmed by restriction digestion, transformed into E. coli HT115 and plated on LB with ampicillin and tetracycline. For RNAi experiments, 12–15 homozygous yc32[gfp::lmn-1]; baf-1(bq19[G12T]) hermaphrodites were transferred at L4 stage to RNAi plates containing dsRNA-producing E. coli HT115, 1 mM IPTG and 100 μg/ml ampicillin and incubated at 20 °C. After 18 h, 1-day old adults were transferred to fresh RNAi plates (3 animals per plate; nine worms in total per replica) and incubated for 6 h at 20 °C. The adults were then removed and rescue of lethality was determined by counting the number of unhatched embryos and viable offspring after 24 h at 20 °C.

**Statistical analysis of *C. elegans* data**. Data were analyzed in R Studio (1.3.1093[71]; running R 4.0.2[72]) and represented with base plotting. Mann-Whitney and t-tests were performed in R Studio and adjusted for multiple comparison (Benjamini & Hochberg method) when relevant. *C. elegans* has two sexes: hermaphrodites and males. The viability experiments reported in this manuscript can only be performed with hermaphrodites. There is therefore no sex specific analysis performed.

### Reporting summary
Further information on research design is available in the Nature Portfolio Reporting Summary linked to this article.

## Data availability
The RNA Seq data has been deposited on GEO with the accession number GSE269484. eQTL data from the eQTLGen study was accessed using the Phase I release of cis-eQTLs and is available for download via https://eqtlgen.org/phase1.html. GWAS summary statistics are available via https://zenodo.org/records/1251813 from the GIANT consortium, via https://csg.sph.umich.edu/willer/public/glgc-lipids2021/ from the GLGC consortium and via http://www.gefos.org/ from the GEFOS consortium. All other source data are provided with the paper. Source data are provided with this paper.

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

## Acknowledgements

We would like to thank Prof. Lopez Otin for providing us with the NGPS and WT immortalized fibroblasts, Dr Daniel Starr for providing us with the *gfp::lmn-1* alleles, Dr Dimitri Shcherbakov and Dr Erik Böttger for sharing the dual luciferase mistranslation assay plasmids. We thank Dr Dimitri Shcherbakov and Dr Mark Stoneley for helpful discussions on setting up and interpreting the assays, Matthew Gratian (CIMR microscopy) for his help with the high content microscope and screen set up, Henri Huppert (ThermoFisher) for his help with the HCS studio pipelines and Dr Gabriel Balmus for giving us access to the CyBio Felix robot. Funding: Sir Henry Dale Fellowship jointly funded by the Wellcome Trust and the Royal Society 206242/Z/17/Z (S.Y.B., D.L.). Wellcome Trust Institutional Strategic Support Fund 204845/Z/16/Z (A.F.J.J., D.L.). FEBS Long-Term Fellowship (A.F.J.J.). Spanish State Research Agency Grants PID2022-137162NB-I00 and CEX2020-001088-M (P.A.). Spanish State Research Agency Fellowship BES-2017-080216 (R.R.B.). Medical Research Council (Unit programs: MC_UU_12015/2, MC_UU_00006/2 (J. P., K.K.). Medical Research Council (MRC) [research grants MR/M010007/1 and MR/R0009015/1 to N.A.B. Swedish Research Council (VR) (2017-06088 and 2019-04868), the Swedish Cancer Society (Cancerfonden) (20 1034 Pj and 23 2994 Pj), and the Novo Nordisk Foundation (NNF21OC0070427 and NNF22OC0078353) to C.G.R.

## Author contributions

Conceptualization: DL. Methodology: S.Y.B., J.H., A.F.J.J., A.F.L., R.R.B., C.G.R., K.K., N.A.B., K.K.O., J.R.B.P., D.L. Investigation: S.Y.B., J.H., A.F.J.J., A.F.L., R.R.B., K.K., N.A.B., D.L. Visualization: S.Y.B., A.F.J.J., A.F.L., R.R.B., K.K., N.A.B., P.A., D.L. Funding acquisition: D.L., P.A., J.R.B.P., K.K.O. Project administration: D.L. Supervision: D.L., P.A., J.R.B.P., K.K.O. Writing—original draft: D.L., S.Y.B. Writing—review & editing: S.Y.B., D.L., J.H., A.F.J.J., K.K., J.R.B.P., P.A., N.A.B., K.K.O. All authors have approved the final version of this manuscript.

## Competing interests

D.L. is an employee of Altos Labs. J.R.B.P. is an employee/shareholder of Insmed. J.R.B.P. also receives research funding from GSK and consultancy fees from WW International. The other authors declare that they have no competing interests.
