## [Peer Review File · Nature Communications]

REVIEWER COMMENTS

Reviewer #1 (Remarks to the Author):

This is an elegant study performed in the lab of Delphine Larrieu in collaboration with other experts in nuclear architecture and function. Breusegem et al carried out a whole genome (~20,000 genes) CRISPR/Cas9 arrayed microscopy screen in Nestor Guillermo Progeria Syndrome patient's fibroblasts (BAF A12T mutation) to identify genes and pathways contribute to aging. The ability of depletion of each gene to rescue aging phenotypes was assessed. The study identified 43 genes in pathways of protein translation and bone cell development as key regulators of aging and progeria. Overall, the study has many strengths and just minor weaknesses.

Strengths:

- An elegant “first genome-wide” multiparametric screen in NGPS cells is presented, with the appropriate controls.
- The monitoring of four aging/progeria phenotypes -nuclear shape, micronuclei, NE ruptures/ blebs and emerin intensity in nucleus/cytoplasm- is a good representation of progeria cellular phenotypes.
- Identified 43 genes that when depleted improve progeria phenotypes, which suggest potential new pathways involved in aging. Of these, 30 genes mapped close to genome-wide signals for traits of skeletal abnormalities and lipodystrophy (interrogating large-scale population GWAS studies).
- The signature showed an enrichment in protein translation genes, which was demonstrated by an increase in protein translation rate and in protein synthesis in NGPS cells, as well as an increase in errors during protein synthesis. These defects were reduced upon depletion of identified hits (21 hits reaching significance). Interestingly, some of these hits are not known to play a role in protein translation, and inhibition of protein synthesis with cycloheximide or silvestrol improved NGPS phenotypes.
- The ultimate proof of the significance of the hits identified in the screen is that depletion of some of them -RPS3A, PAFAH1B1, VPS16, SMU1- suppresses the larvae lethality of a progeria *C. elegans* model (baf -1 mutation in a background of *lmn-1* deficiency).
- Other signatures identified in NGPS include protein and RNA transport and osteoclast development, but these are not investigated in the study.
- Overall, the results support the author's working model and suggest that increased protein synthesis and translation errors could contribute to aging phenotypes in NGPS. In addition, the study reveals an effect on protein synthesis of proteins not known to participate in this process. Moreover, many of the hits identified could exert anti-aging effects independently of protein

synthesis. The significance of the current study is that provides new potential NGPS pathways that could contribute to other progerias and normal aging.

Weaknesses and minor comments:

- Protein translation is reduced significantly with respect to WT cells upon correction of BAF A12T mutation in Fig 5A. How do the authors explain this?
- Nucleolar area of WT cells is not statistically different from NGPS1 or NGPS2 cells (Fig S5D), even though the NGPS2 corrected cells exhibit reduced nucleolar area (Fig 1D). Based on this, it is not clear whether the nucleolar area means anything.
- The mistranslation assay does not include data on the NGPS2 corrected sample.
- The authors use in their screen immortalized NGPS cells via SV40LT and TERT. Both inactivation of p53/pRb pathways (SV40LT) and expression of telomerase (TERT) are known to impact senescence and aging phenotypes. Telomerase in particular, can rescue many aging hallmarks in other progerias. This needs to be discussed in the manuscript because the results in primary NGPS cells could be different.

Reviewer #2 (Remarks to the Author):

In this manuscript, Breusegem and his colleagues performed a CRISPR/Cas9 arrayed microscopy screen in fibroblasts derived from Néstor-Guillermo progeria syndrome (NGPS) patients. The screening was designed to identify potential interventions capable of reversing the observed abnormal phenotypes in NGPS fibroblasts, including delocalization of emerin into cytoplasm, nuclear deformation, nuclear envelope ruptures and increased micronuclei formation. The authors pinpointed 43 candidate genes of interest through screening, and further validated the function of protein synthesis-associated genes by employing siRNA-mediated knockdown. This functional validation confirmed that depletion of hits involved in protein translation ameliorated the disorganization of nuclear envelope. While these findings are interesting, several critical points must be addressed to ensure the logical clarity and reproducibility of the experiments, particularly regarding the connection between the abnormal phenotypes observed in NGPS fibroblasts and the premature aging phenotype of the patient, and the necessity for additional experiments aimed at elucidating the underlying mechanism through which these genes influence nuclear morphology and ameliorate the abnormal phenotypes of NGPS. Authors also conducted an in vivo investigation involving the inhibition of 32 homologous genes in *C. elegans* models, showcasing a rescue of lethality. While this presents a crucial aspect of their study, it is important to acknowledge that there are substantial concerns regarding the chosen model organism, the underneath rationale and the data analysis.

Below, the specific issues that warrant attention are outlined.

1. The manuscript is titled “A multiparametric anti-aging CRISPR screen...”, yet there is a lack of clarity regarding the rationale behind this title. Specifically, it is unclear whether NGPS fibroblasts undergo any of the events related to aging, such as cell death, senescence or cell cycle arrest? If these events do occur, it is important to determine whether the abnormal phenotypes, such as delocalization of emerin into cytoplasm, nuclear deformation, nuclear envelope ruptures, increased micronuclei formation and the increased protein synthesis, are the driving forces to these aging events. Therefore, conducting experimental investigation to establish this connection is imperative, as it forms a crucial foundation for the overall rationale behind this screening and the following studies.

2. The CRISPR screen in this study appears to lack data reproducibility. To improve the credibility of this dataset, it is essential to include an additional cell line for the same screening. In this study, the CRISPR screen data was generated using NGPS2-Cas9 cell line. Notably, as their data exhibited, the NGPS1 cell line displayed more pronounced nuclear defects and higher P21 expression. Why did the authors choose NGPS2-Cas9 instead of NGPS1 cell line for further analysis?

3. The positive control such as ribosomal proteins (e.g., RPL12, RPL37A, RPL13), as well as some hits genes were validated to reduce the nucleolar area and abolish the nascent protein synthesis in NGPS cells, indicating their potential involvement in ribosome biogenesis. However, there has been no subsequent mechanistic exploration to elucidate how the candidate hits, such as PAFAH1B1, SMU1 and VPS16, which were subjected to in vivo *C. elegans* lethality assay, are intricately involved in this process. Most importantly, it is crucial to investigate their functional role in the cellular phenotypes related to aging (a task yet to be addressed by the authors).

4. Authors employed *gfp::lmn-1; baf-1(G12T)* worms to illustrate that deletion of 4 of 32 *C. elegans* homologue genes rescued the larval lethality. However, they did not provide direct evidence that the worms exhibit an accelerated aging, or that knockdown of any of the candidate hits could extend the lifespan. It is worth noting that the larvae lethality could be due to developmental defects, rendering it unsuitable for aging evaluation. Authors should engage in a more comprehensive discussion regarding the relevance of NGPS cell models and the *C. elegans* model. It is essential to furnish additional information that substantiates the selection of the *C. elegans* model and its applicability to NGPS.

5. Authors should offer more additional information on the mechanism underlying the increase in protein synthesis and translation errors in NGPS cells (Figure 5). This would help elucidate the connection between these phenomena and the symptoms of premature aging.

6. In Figure 6B, it appears that NGPS worms exhibit a smaller nucleolar area compared to WT worms at 16 °C, which contradicts to the observations in human fibroblasts with BAF A12T mutation. The discrepancy needs reasonable explanation, and again, this also needs to align with the underneath rationale for the connection between nucleolar area etc. and aging, as the foundation of this work. Additionally, the temperature chosen for the in vivo experiments was not clearly described in the manuscript.

7. In the Discussion section (line 270), it is mentioned that the knockout of LRRK1 and PAFAH1B1/LIS1 improves NGPS cell phenotypes and enhances NGPS worm survival, but the source of this data is not specified. The discussion provides some insights into the implications of the findings, such as potential therapeutic targets. However, it would be helpful to include a section discussing the study's limitations and avenues for further research.

General issues:

1. Regarding western blotting data, there are concerns about the choice of tubulin and actin as loading control for H3K9me3 and H3K79me3 (e.g., Figure 1I). Additionally, it is noted that molecular weight markers have not been labeled for all western blot analyses (e.g., Figure 1I); In some instances, the brand labels are not clear (e.g., Figure 2J, ac-tubulin) or the brands appear incomplete (e.g., Figure S3F). Lastly, it is recommended to annotate the protein name instead of gene name (e.g., Figure S3H).

2. In Figure 1C, it is questionable to locate Lamin B1 in WT and NGPS cells. The green signals for Lamin B1 in the merged panels do not appear to be located around the nuclear envelop.

3. Figure 1D, it seems like that the 53BP1 signals between WT and NGPS cells are comparable.

4. The conclusions could not be addressed without statistical analysis (error bars and P values) in Figure 2K, S3I and S3K.

5. Gene names should be in italicized format (e.g., Line 217).

6. The following figures lack statistical graphs: Figure 1A, Figure 1C, Figure 1I, Figure 2B, Figure 4C.

Reviewer #3 (Remarks to the Author):

This manuscript by Breusegem et al describes a throughput screening method of identifying new players in NGPS. Using parameters such as nuclear morphology, they implicate previously unknown proteins in the phenotype displayed by immortalised NGPS cells and in a *C. Elegans*

model of the disease. Despite the interesting data presented, the authors fail to validate the importance in primary NGPS cells (non immortalised) and a number of critical experiments are only 2 independent experiments. Please see below a list of my concerns:

- 1) Several experiments are only 2 independent replicates. These need to be increased to $n = 3$.
- 2) It is better to plot graphs to show the means of the independent replicates, rather than the spread of every data point. Was the statistical testing performed on the means of the replicates?
- 3) Some figure legends do not contain information of the n numbers for graphs presenting data.
- 4) Suggest a statistical review as I believe that t -tests are not the correct statistical analysis. On graphs presenting multiple conditions, one-way anova should be performed (eg Fig 1 E and F etc).
- 5) Figure 6B – Should be analysed by 2-way anova and is only an $n=2$ independent experiments.
- 6) Figure 2E – can nuclear envelope rupture be better characterised? Do blebs always lead to rupture? Better images and analysis are required. The NGPS cells have normal heterochromatin levels, is this a problem with nuclear envelope coupling? More mechanistic data would increase the interest in the manuscript.
- 7) NGPS, like other progeria syndromes, is a disease of premature ageing and senescence. While I can understand immortalised cells being used for the screening, further validation should be performed on the primary cells. Or maybe on iPSCs? This would further support the authors findings and increased the importance/interest in this manuscript.
- 8) Does BAF directly play a role in protein synthesis? More mechanistic data is required here would increase the interest in the manuscript.

REVIEWER

COMMENTS

Reviewer #1 (Remarks to the Author):

This is an elegant study performed in the lab of Delphine Larrieu in collaboration with other experts in nuclear architecture and function. Breusegem et al carried out a whole genome (~20,000 genes) CRISPR/Cas9 arrayed microscopy screen in Nestor Guillermo Progeria Syndrome patient's fibroblasts (BAF A12T mutation) to identify genes and pathways contribute to aging. The ability of depletion of each gene to rescue aging phenotypes was assessed. The study identified 43 genes in pathways of protein translation and bone cell development as key regulators of aging and progeria. Overall, the study has many strengths and just minor weaknesses.

Strengths:

- An elegant “first genome-wide” multiparametric screen in NGPS cells is presented, with the appropriate controls.
- The monitoring of four aging/progeria phenotypes -nuclear shape, micronuclei, NE ruptures/ blebs and emerin intensity in nucleus/cytoplasm- is a good representation of progeria cellular phenotypes.
- Identified 43 genes that when depleted improve progeria phenotypes, which suggest potential new pathways involved in aging. Of these, 30 genes mapped close to genome-wide signals for traits of skeletal abnormalities and lipodystrophy (interrogating large-scale population GWAS studies).
- The signature showed an enrichment in protein translation genes, which was demonstrated by an increase in protein translation rate and in protein synthesis in NGPS cells, as well as an increase in errors during protein synthesis. These defects were reduced upon depletion of identified hits (21 hits reaching significance). Interestingly, some of these hits are not known to play a role in protein translation, and inhibition of protein synthesis with cycloheximide or silvestrol improved NGPS phenotypes.
- The ultimate proof of the significance of the hits identified in the screen is that depletion of some of them -RPS3A, PAFAH1B1, VPS16, SMU1- suppresses the larvae lethality of a progeria *C. elegans* model (baf -1 mutation in a background of *lmm-1* deficiency).
- Other signatures identified in NGPS include protein and RNA transport and osteoclast development, but these are not investigated in the study.
- Overall, the results support the author's working model and suggest that increased protein synthesis and translation errors could contribute to aging phenotypes in NGPS. In addition, the study reveals an effect on protein synthesis of proteins not known to participate in this process. Moreover, many of the hits identified could exert anti-aging effects independently of protein synthesis. The significance of the current study is that provides new potential NGPS pathways that could contribute to other progerias and normal aging.

Weaknesses and minor comments:

- Protein translation is reduced significantly with respect to WT cells upon correction of BAF A12T mutation in Fig 5A. How do the authors explain this?

While there is indeed a statistically significant difference between the WT and the NGPS2 corrected cells (see updated Figure 5C; statistical comparison using one-way ANOVA with Tukey's multiple comparison testing), we believe that the NGPS2 "corrected" cell line is a better control for comparison with the NGPS2 cells as it is an isogenic line engineered from the NGPS2 cells; the different genetic background of the WT cells could explain the difference with the NGPS2 corrected cells.

- Nucleolar area of WT cells is not statistically different from NGPS1 or NGPS2 cells (Fig S5D), even though the NGPS2 corrected cells exhibit reduced nucleolar area (Fig 1D). Based on this, it is not clear whether the nucleolar area means anything.

It is true that even though there is a trend in the average nucleolar area being larger in NGPS1 and 2 compared to WT, the difference is not statistically significant. All we can say is that indeed, when comparing isogenic cell lines (NGPS2 and corrected line), the nucleolar area is going down. We have now modified the text to clarify that: "However, even though we observed a modest but significant increase in the nucleolar area of NGPS2 cells compared to NGPS2 corrected cells (Fig. 5E-F), as well as in fibroblasts expressing BAF A12T compared to BAF WT (Fig. 5G), no significant difference was observed between the WT cells and the NGPS cells (Fig. S5C, D)." (page 8)

- The mistranslation assay does not include data on the NGPS2 corrected sample.

We did perform the mistranslation assay on the NGPS2 corrected cells; however, their readthrough was not decreased compared to the NGPS2 cells (see left graph below, Student's *t*-test). While we can't readily explain this result (could it be that the editing added translation stress to the cells, or could the difficulty in having similar cell confluency with the 2 cell lines be a factor?), we did observe a small but significant increased readthrough in WT cells overexpressing BAF A12T (see right graph below, one-way ANOVA with Dunnett's multiple comparison testing), which further corroborates the effect of BAF A12T expression on protein translation errors.

- The authors use in their screen immortalized NGPS cells via SV40LT and TERT. Both inactivation of p53/pRb pathways (SV40LT) and expression of telomerase (TERT) are known

to impact senescence and aging phenotypes. Telomerase in particular, can rescue many aging hallmarks in other progerias. This needs to be discussed in the manuscript because the results in primary NGPS cells could be different.

We acknowledge this comment, and agree with the reviewer that being able to perform some experiments in primary cells would have been the best. However, cells were only obtained from two out of the three so far identified NGPS patients, and the primary cells could not be maintained in culture as they did not proliferate (personal communication from Dr. Carlos Lopez-Otin). Therefore, these cells do not exist anymore, and only the immortalised cells are now available for research use. We now explain the use of these cells better in a new paragraph on p.3 of the manuscript: “These are the only NGPS cells available to the scientific community, as non-immortalized primary cells were found to be unable to proliferate in culture (personal communication from Dr. Carlos Lopez-Otin).”

However, the recent BioRxiv paper deposited by the group of Peter Askjaer (<https://doi.org/10.1101/2024.03.17.585430>), who is a co-author on this paper, has established the phenotypes of the NGPS *C. elegans*. The worms display shorter lifespan, decreased resistance to stress, and importantly similar phenotypes as the ones we observed in NGPS cells: nuclear abnormalities, loss of emerin at the nuclear envelope, nucleolar enlargement and differential expression of genes regulating ribosomes and translation. Therefore, we believe that our validation studies in this *in vivo* NGPS model strongly support our results in immortalized NGPS cells.

Reviewer #2 (Remarks to the Author): In this manuscript, Breusegem and his colleagues performed a CRISPR/Cas9 arrayed microscopy screen in fibroblasts derived from Néstor-Guillermo progeria syndrome (NGPS) patients. The screening was designed to identify potential interventions capable of reversing the observed abnormal phenotypes in NGPS fibroblasts, including delocalization of emerin into cytoplasm, nuclear deformation, nuclear envelope ruptures and increased micronuclei formation. The authors pinpointed 43 candidate genes of interest through screening, and further validated the function of protein synthesis-associated genes by employing siRNA-mediated knockdown. This functional validation confirmed that depletion of hits involved in protein translation ameliorated the disorganization of nuclear envelope. While these findings are interesting, several critical points must be addressed to ensure the logical clarity and reproducibility of the experiments, particularly regarding the connection between the abnormal phenotypes observed in NGPS fibroblasts and the premature aging phenotype of the patient, and the necessity for additional experiments aimed at elucidating the underlying mechanism through which these genes influence nuclear morphology and ameliorate the abnormal phenotypes of NGPS. Authors also conducted an *in vivo* investigation involving the inhibition of 32 homologous genes in *C. elegans* models, showcasing a rescue of lethality. While this presents a crucial aspect of their study, it is important to acknowledge that there are substantial concerns regarding the chosen model organism, the underneath rationale and the data analysis.

Below, the specific issues that warrant attention are outlined. 1. The manuscript is titled “A multiparametric anti-aging CRISPR screen...”, yet there is a lack of clarity regarding the rationale behind this title. Specifically, it is unclear whether NGPS fibroblasts undergo any of the events related to aging, such as cell death, senescence or cell cycle arrest? If these events do occur, it is important to determine whether the abnormal phenotypes, such as delocalization of emerin into cytoplasm, nuclear deformation, nuclear envelope ruptures, increased micronuclei formation and the increased protein synthesis, are the driving forces to these aging events. Therefore, conducting experimental investigation to establish this connection is

imperative, as it forms a crucial foundation for the overall rationale behind this screening and the following studies.

We acknowledge this comment, and agree with the reviewer that being able to perform some experiments in primary cells would have allowed us to address some of these questions. However, cells were only obtained from two out of the three so far identified NGPS patients, and the primary cells could not be maintained in culture as they did not proliferate (personal communication from Dr. Carlos Lopez-Otin). Therefore, these cells do not exist anymore, and only the immortalised cells are now available for research use.

We now explain the use of these cells better in a new paragraph on p.3 of the manuscript: “These are the only NGPS cells available to the scientific community, as non-immortalized primary cells were found to be unable to proliferate in culture (personal communication from Dr. Carlos Lopez-Otin).”

This prevents the feasibility of “cellular ageing” assays. However, as described in the response to reviewer 1, a manuscript recently deposited on BioRxiv (<https://doi.org/10.1101/2024.03.17.585430>) from the lab of Dr Peter Askjaer, (one of the co-author on the paper) now strongly supports our cellular data. The manuscript shows that the *C. elegans* NGPS model presents nuclear morphology deterioration, and deregulation of ribosome components, associated with a significantly reduced lifespan. We believe that this is the best alternative model available to confirm the ageing phenotypes associated with NGPS.

2.The CRISPR screen in this study appears to lack data reproducibility. To improve the credibility of this dataset, it is essential to include an additional cell line for the same screening. In this study, the CRISPR screen data was generated using NGPS2-Cas9 cell line. Notably, as their data exhibited, the NGPS1 cell line displayed more pronounced nuclear defects and higher P21 expression. Why did the authors choose NGPS2-Cas9 instead of NGPS1 cell line for further analysis?

We would like to thank the reviewer for this comment but would like to give more insights into the screening process. We are not sure what the reviewer is referring to when mentioning lack of data reproducibility.

Running a whole genome arrayed screen requires a huge amount of resources, not only in terms of cost but also in terms of time – especially when only ran by one person in the lab. Such primary screens require the use of 61 x 384 well plates and involves many steps as schematised on Fig. 3A: reverse transfection of the cells, fixation, primary antibodies incubation, washes, secondary antibodies incubation, washes and then of course high content microscopy imaging in 4 channels, and analysis of the images in the four colours for all of these plates. For this reason, such screens are always only performed once and in one cell line, before the hits can then be validated using triplicate repeats (see previous literature doing whole genome siRNA screens for example: <https://doi.org/10.1038/s41597-021-00944-5> or <https://doi.org/10.15252/emboj.201487826>). This is what we have done in this paper, and we currently don't have the resources to repeat such a massive screen in the second NGPS cell line. We hope the reviewer will understand the reasoning behind that.

The reason for choosing NGPS2 to run the screen is two fold: first, the nuclei of the NGPS2 cells were bigger than the ones in the NGPS1 cells (see examples in Fig. S1) and therefore more amenable to image-based screening for the phenotypes we were looking at. Secondly, we only managed to correct the NGPS mutation in the NGPS2 cell line, which allowed us to

establish isogenic cell lines to validate the screening phenotypes (Fig. S2) and we thought this was an important part of the screen set up.

We have clarified this by adding a sentence in p.4 of the manuscript: “To this aim, we took advantage of our recently established NGPS2 isogenic cell lines, in which we reversed the homozygous BAF A12T mutation using CRISPR/Cas9 (NGPS2 corrected) (14). For this reason, we carried out further validation using the NGPS2 cell line.”

3. The positive control such as ribosomal proteins (e.g., RPL12, RPL37A, RPL13), as well as some hits genes were validated to reduce the nucleolar area and abolish the nascent protein synthesis in NGPS cells, indicating their potential involvement in ribosome biogenesis. However, there has been no subsequent mechanistic exploration to elucidate how the candidate hits, such as PAFAH1B1, SMU1 and VPS16, which were subjected to in vivo *C. elegans* lethality assay, are intricately involved in this process. Most importantly, it is crucial to investigate their functional role in the cellular phenotypes related to aging (a task yet to be addressed by the authors).

We don't have any data on the functional role of these genes. Nevertheless, our observation that depletion of proteins related to PAFAH1B1/LIS1 and VPS16, notably DYNLL2, VPS11 (and others) also suppress the lethality supports the specificity of the identified pathways. As discussed in the manuscript, genes implicated in lysosomal/vesicles transport and osteoclast homeostasis are likely to be relevant for NGPS progression. We feel that the elucidation of the function of these genes in *C. elegans* is beyond the scope of this manuscript as this could be the focus of a full independent study.

4. Authors employed *gfp::lmn-1*; *baf-1(G12T)* worms to illustrate that deletion of 4 of 32 *C. elegans* homologue genes rescued the larval lethality. However, they did not provide direct evidence that the worms exhibit an accelerated aging, or that knockdown of any of the candidate hits could extend the lifespan. It is worth noting that the larvae lethality could be due to developmental defects, rendering it unsuitable for aging evaluation. Authors should engage in a more comprehensive discussion regarding the relevance of NGPS cell models and the *C. elegans* model. It is essential to furnish additional information that substantiates the selection of the *C. elegans* model and its applicability to NGPS.

C. elegans is a highly appreciated model to study mechanisms of ageing. Numerous studies have concluded that interventions that either extend or shorten lifespan in worms have similar

effects in other animals, including mammals (doi 10.1016/j.cell.2005.02.002; 10.1038/s41467-023-35869-7; 10.3389/fendo.2020.554994). Secondly, the strong conservation of BAF across the animal kingdom both in terms of primary amino acid sequence and predicted secondary structure (see figure below) argues in favour of functional conservation. For instance, we and others have shown that phosphorylation of BAF by the kinase VRK1 regulates BAF dynamics similarly in worms, flies, and human cells.

As we mentioned in the response to the first comment of the reviewer, the manuscript recently deposited on BioRxiv from the lab of Dr Peter Askjaer, now describes the phenotypes of the worms, that display shorter lifespan, nuclear and NE abnormalities and deregulation of genes associated with translation. This now strongly supports that *C. elegans* is a good model to study NGPS, as well as strengthens our data obtained in cells. We have now added a paragraph in the discussion section to explain the use of the *C. elegans* model further (p.11)

5. Authors should offer more additional information on the mechanism underlying the increase in protein synthesis and translation errors in NGPS cells (Figure 5). This would help elucidate the connection between these phenomena and the symptoms of premature aging. We would like to thank the reviewer for this comment. To address this, we have performed RNA seq analysis in NGPS 1 and 2 cell lines, compared to WT. We observed 213 genes that were being deregulated in NGPS cells, and interestingly, GO analysis highlighted a strong enrichment for genes involved in translation regulation (See new figure panels 5A-B). The direct role of BAF in modulating the expression of translation-related genes, is also supported by the Biorxiv data on NGPS *C. elegans*, showing that BAF A12T DNA binding profile is modified, compared to BAF WT, leading to differential gene expression, especially for ribosomal genes. Together, we feel that these new results address the reviewer's comment and provide more mechanistic insights into how BAF A12T modulates RNA translation.

6. In Figure 6B, it appears that NGPS worms exhibit a smaller nucleolar area compared to WT worms at 16 °C, which contradicts to the observations in human fibroblasts with BAF A12T mutation. The discrepancy needs reasonable explanation, and again, this also needs to align with the underneath rationale for the connection between nucleolar area etc. and

aging, as the foundation of this work. Additionally, the temperature chosen for the in vivo experiments was not clearly described in the manuscript.

Indeed, we don't really have an explanation for the differences observed at 16C and 25C. In the revised graph in our manuscript, we have plotted the same data again here obtained from more animals and omitting the 16C data; . The p value is 0.033 so it reaches significance (New figure 7A-B)

7. In the Discussion section (line 270), it is mentioned that the knockout of LRRK1 and PAFAH1B1/LIS1 improves NGPS cell phenotypes and enhances NGPS worm survival, but the source of this data is not specified.

The corresponding figures have now been indicated in the text.

The discussion provides some insights into the implications of the findings, such as potential therapeutic targets. However, it would be helpful to include a section discussing the study's limitations and avenues for further research.

We have now added a couple of new paragraphs across the discussion, to further discuss the limitations of the study and future avenues of research.

General issues:

1. Regarding western blotting data, there are concerns about the choice of tubulin and actin as loading control for H3K9me3 and H3K79me3 (e.g., Figure 1I). Additionally, it is noted that molecular weight markers have not been labeled for all western blot analyses (e.g., Figure 1I); In some instances, the brand labels are not clear (e.g., Figure 2J, ac-tubulin) or the brands appear incomplete (e.g., Figure S3F). Lastly, it is recommended to annotate the protein name instead of gene name (e.g., Figure S3H).

Western blots for H3K9me3 and H3K79me2 were repeated with 3 independent sets of lysates and now quantified using histone H3 as loading control. Molecular weight markers were added to the Western blots shown and band labels corrected as suggested.

2. In Figure 1C, it is questionable to locate Lamin B1 in WT and NGPS cells. The green signals for Lamin B1 in the merged panels do not appear to be located around the nuclear envelop.

In the merged panels (right-most panels of Figure 1B) overlap of the emerin (pseudocoloured magenta) and lamin B1 (pseudocoloured green) signals is indicated by white; therefore a white nuclear envelope in the merged panel does indicate colocalization of emerin and lamin B1 at the nuclear envelope. We hope this clarifies the figure for the reviewer.

3. Figure 1D, it seems like that the 53BP1 signals between WT and NGPS cells are comparable. Yes, the signals are comparable intensity-wise; we are however quantifying in Figure 1D the number of 53BP1 puncta, not their intensity.

4. The conclusions could not be addressed without statistical analysis (error bars and P values) in Figure 2K, S3I and S3K.

We have now included biological repeats and statistical analysis.

5. Gene names should be in italicized format (e.g., Line 217).

Gene names have now been italicized.

6. The following figures lack statistical graphs: Figure 1A, Figure 1C, Figure 1I, Figure 2B, Figure 4C.

We have added quantitation for the Western blots in Figure 1. For Figure 2B, the quantitation is in Figs 2C/D/H/I. For Figure 4C all image parameters that were quantified are listed in Table S7. We did not quantify the EM observations (as represented in Figure 1A) or the emerin/lamin B1 co-localization (as shown in Figure 1B). The images shown are representative, are only discussed qualitatively in the text and do not bear further on the screening results.

Reviewer #3 (Remarks to the Author):

This manuscript by Breusegem et al describes a throughput screening method of identifying new players in NGPS. Using parameters such as nuclear morphology, they implicate previously unknown proteins in the phenotype displayed by immortalised NGPS cells and in a C. Elegans model of the disease. Despite the interesting data presented, the authors fail to validate the importance in primary NGPS cells (non immortalised) and a number of critical experiments are only 2 independent experiments. Please see below a list of my concerns:

1) Several experiments are only 2 independent replicates. These need to be increased to $n = 3$. Where possible we have increased the number of independent replicates to $n = 3$ (please see figure legends).

2) It is better to plot graphs to show the means of the independent replicates, rather than the spread of every data point. Was the statistical testing performed on the means of the replicates? Yes, the statistical testing was performed on the means of the replicates. The spread of the data points is shown in some graphs only (Figs 5A/B/D/E/updated panel F) as “SuperPlots”, where a larger symbol represents the mean of the data points in a particular replicate, and these mean values were used for the statistical comparisons. Furthermore the mean of the means of the replicates is indicated by a line. We can replace the Superplots with bar graphs if this is preferred by the reviewers/editor.

3) Some figure legends do not contain information of the n numbers for graphs presenting data. We have now included the number of replicates in the figure legends.

4) Suggest a statistical review as I believe that t-tests are not the correct statistical analysis. On graphs presenting multiple conditions, one-way anova should be performed (eg Fig 1 E and F etc).

We would like to thank the reviewer and have now performed a thorough statistical review. We re-analysed most of the data using one-way ANOVA; since we are comparing the 2 NGPS cell lines to the same control cell line (or, in Figure 6B, 2 drug treatments to untreated cells), we have used Dunnett’s multiple comparison testing and have used the multiplicity-adjusted p -values to determine significance. In Figure S3 and Figure 5C the 3 different cell lines were compared using one-way ANOVA with Tukey’s multiple comparison testing.

5) Figure 6B – Should be analysed by 2-way anova and is only an $n=2$ independent experiments.

This now appears as the new Figure 7B. It has now been repeated in 4 independent experiments, and the Welch two sample t test has been used as described in the figure legend.

6) Figure 2E – can nuclear envelope rupture be better characterised? Do blebs always lead to

rupture? Better images and analysis are required. The NGPS cells have normal heterochromatin levels, is this a problem with nuclear envelope coupling? More mechanistic data would increase the interest in the manuscript.

We thank the reviewer for this comment. In fact, a paper from our lab was published last year in *Nucleic Acids research*, studying and characterising the ruptures in NGPS cells in a lot more details. We would like to refer the reviewer to this manuscript: [10.1093/nar/gkac726](https://doi.org/10.1093/nar/gkac726)

In that study, we showed that the BAF A12T mutation in NGPS cells affects the binding affinity between BAF and Lamin A/C, preventing the recruitment of Lamin A/C to the sites of NE ruptures, leading to frequent NE re-rupturing in interphase cells – we have added this paragraph into the main text now.

NGPS cells seem to actually have similar levels of specific heterochromatin markers such as HP1 gamma (Fig. 1H-I). However, we never observed defects in the LINC complex or in the cytoskeleton organisation in these cells (data not shown), on the contrary to what has been observed in HGPS cells. This suggests that there might not be an obvious issue in nucleocytoskeleton coupling in NGPS cells but this could be the scope of a separate study which we feel is outside the remit of the current manuscript.

7) NGPS, like other progeria syndromes, is a diseased of premature ageing and senescence. While I can understand immortalised cells being used for the screening, further validation should be performed on the primary cells. Or maybe on iPSCs? This would further support the authors findings and increased the importance/interest in this manuscript.

Please see the response to reviewers 1 and 2, regarding the recently deposited Biorxiv paper describing the NGPS *C. elegans* model.

8) Does BAF directly play a role in protein synthesis? More mechanistic data is required here would increase the interest in the manuscript.

We would like to thank the reviewer for this comment. To address this, we have performed RNA seq analysis in NGPS 1 and 2 cell lines, compared to WT. We observed 213 genes that were being deregulated in NGPS cells, and interestingly, GO analysis highlighted a strong enrichment for genes involved in translation regulation (See new figure panels 5A-B). The direct role of BAF in modulating the expression of translation-related genes, is also supported by the BioRxiv data on NGPS *C. elegans*, showing that BAF A12T DNA binding profile is modified, compared to BAF WT, leading to differential gene expression, especially for ribosomal genes. Together, we feel that these new results address the reviewer's comment and provide more mechanistic insights into how BAF A12T modulates RNA translation.

REVIEWER COMMENTS

Reviewer #1 (Remarks to the Author):

The authors have addressed in the text the concerns expressed by reviewers.

Reviewer #2 (Remarks to the Author):

Breusegem et al. have introduced some improvements, however, the primary concerns I had remain unaddressed in the current manuscript.

It is well-recognized that senescent cells typically exhibit reduced global protein synthesis or translation. Therefore, it is unclear whether immortalized NPG fibroblasts are suitable for anti-aging CRISPR screening, given that their overall translation may be less disrupted. The authors should substantiate their conclusion using SUNSET analysis or polysome profiling experiments.

Despite performing some work (e.g., RNA-seq, HPG labeling) and citing a preprint manuscript to suggest a role for BAF in protein synthesis, there is no conclusive data demonstrating that BAF is essential for regulating protein synthesis, nucleolar area, and cellular senescence.

The protein synthesis rate in *C. elegans* should also be measured following the deletion of these targets (RPS3A, PAFAH1B1, VPS16 and SMU1).

In Figures 4C-4G and 5B, the size and color should be annotated for clarity.

Reviewer #3 (Remarks to the Author):

The authors have addressed all my comments in this revised manuscript. The manuscript presents interesting, novel findings that are statistically sound. This paper will be of great interest to the field and the journal readership.

I recommend accept.

Reviewer #4 (Remarks to the Author):

In short, NCOMMS-23-35016A describes genetic screening of gene deficiencies that ameliorate cellular phenotypes associated with BAF A12T mutants. BAF A12T is associated with Guillermo Progeria Syndrome (NGPS), a progeric disease associated with defects in the nuclear envelope (e.g., lamin or baf mutations). This genetic approach, e.g., looking for ‘suppressor mutants’ of disease phenotypes, is timely and important and provides an inroad for defining drug/drug targets benefiting affected patients in the long term. To the best of my knowledge, I am not in the inner circle of the field; a corresponding knockout mouse has not been described. Thus, the authors cannot expect to provide a mouse model or mouse double mutants as part of their study (showing rescue of progeric phenotypes). I understand that the revision process of this paper is already taking well over a year.

Of course, using *C elegans* is a very valid nonvertebrate model for aging and a preprint from one or two of the coauthor’s labs <https://doi.org/10.1101/2024.03.17.585430> shows that the nematode BAF A12T mutants have a modestly shortened life span compared to wild-type, the phenotype being enhanced in nematodes not carrying a germ line. Besides aging, these worms also show reduced fecundity when aging and other phenotypes are associated with nuclear organization. Thus, it would, in principle, be possible to extend the nematode analysis; this, however, would compromise the novelty of the detailed study described in the preprint. As a reminder, in NCOMMS-23-35016A, the authors use a most valid but maybe more indirect assay to test phenotypes associated with nematode BAF A12T phenotypic suppression: A double mutant with lamin and baf A12T, which arrests in the L1 stage is used, and the suppression of this phenotype is shown.

All in all, I suggest a two-pronged approach.

1) I think the abstract and possible introduction and discussion need to be further tuned down,

eg. To be more specific in stating that multiple cellular phenotypes associated with progeria could be rescued by Rather than ‘identifying 43 new genes that can reverse multiple aging phenotypes in progeria.’

2) I suggest that the editor and key authors have a conversation as to the status of <https://doi.org/10.1101/2024.03.17.585430>. Ideally, I would see a) suppression of direct aging-associated phenotypes (e.g., reduced life span, reduced fecundity, etc.) or b) a better explain the

approach of using genetically sensitized nematodes (e.g., *lmn-1* mutant background). This could be easily done by RNAi, but I understand that the authors might be hesitant to compromise the novelty of their study, which may still have to be accepted for publication.

General Note. I suggest not using the 'whole genome synthetic rescue screens', e.g., avoid 'synthetic rescue screen,' which, to my best knowledge, is not used in genetics terminology. E.g., say, a 'genome-wide suppressor screen' a 'a genome-wide screen for genetic suppressors', 'a genome-wide screen for suppressing and bypassing progeric phenotypes' associated with A12T Guillermo Progeria Syndrome (NGPS) BAF A12T cell line/.

Point-by-point response to the reviewers.

Reviewer #2 (Remarks to the Author)

Breusegem et al. have introduced some improvements, however, the primary concerns I had remain unaddressed in the current manuscript.

It is well-recognized that senescent cells typically exhibit reduced global protein synthesis or translation. Therefore, it is unclear whether immortalized NPG fibroblasts are suitable for anti-aging CRISPR screening, given that their overall translation may be less disrupted. The authors should substantiate their conclusion using SUnSET analysis or polysome profiling experiments.

Despite performing some work (e.g., RNA-seq, HPG labeling) and citing a preprint manuscript to suggest a role for BAF in protein synthesis, there is no conclusive data demonstrating that BAF is essential for regulating protein synthesis, nucleolar area, and cellular senescence.

The protein synthesis rate in *C. elegans* should also be measured following the deletion of these targets (RPS3A, PAFAH1B1, VPS16 and SMU1).

In Figures 4C-4G and 5B, the size and color should be annotated for clarity.

I appreciate this reviewer's comments, however I feel that the concern about senescent cells is not relevant. Indeed, the reviewer mentions that senescent cells have reduced protein synthesis, which has indeed been shown previously.

However, progeria cells appear to be different in that sense. Indeed, Buchwalter and Hetzer previously showed that PRIMARY HGPS fibroblasts also display increased protein synthesis: <https://doi.org/10.1038/s41467-017-00322-z>

Therefore, I don't believe that the increase protein synthesis we see in immortalised NGPS cells is irrelevant. In any case, even if primary NGPS cells would have been available, the cells would have been used when they were still in a proliferative state, not in a senescent state. We are not trying to connect any phenotypes to senescence in this paper, but rather to nuclear envelope dysfunction that we know occur way upstream of senescence.

To address the point of the reviewer, we have added a paragraph in the discussion section of the manuscript to discuss the link between protein synthesis regulation and senescence (line 361):

"It therefore appears that fibroblasts derived from both HGPS and NGPS patients go through a phase of enhanced protein synthesis while they are still in a replicative state (low passage number in the case of primary HGPS cells or immortalization in the case of NGPS). As translation is one of the most energy consuming process in the cell, this could accelerate cell exhaustion, contributing to premature entry of the cells into senescence. Once the cells enter into senescence, they then display reduced global protein synthesis. As the NGPS cells are immortalized, we cannot assess the effect of protein synthesis reduction on the replicative lifespan of the cells in culture, and therefore we can't connect the phenotypic rescue to cellular senescence."

Substantiating our data with SUnSET analysis or polysome profiling would not address the reviewer's concern, as this would still be done on the same immortalised NGPS cells.

However, we did perform some polysome profiling experiment that showed a very small increase of heavy polysomes in NGPS87 and a reduction of polysomes in the corrected NGPS87 cells. As the difference is quite small (but reproducible), and after discussing with the editor, we did not include the data in the manuscript.

We have checked everywhere in the manuscript and made sure that there is no claim that BAF is essential for protein synthesis. Instead, what we suggest is that BAF plays a role in regulating the rate and the fidelity of protein synthesis. We do believe that the preprint paper from the lab of Peter Askjaer strongly supports this conclusion.

We have now toned down the claims relating BAF and protein synthesis regulation across the manuscript. For e.g. in several instances instead of saying that the BAF A12T mutation causes protein synthesis dysfunction we now say that it is associated with protein synthesis deregulation (please see track changes across the manuscript).

Reviewer #4 (Remarks to the Author)

In short, NCOMMS-23-35016A describes genetic screening of gene deficiencies that ameliorate cellular phenotypes associated with BAF A12T mutants. BAF A12T is associated with Guillermo Progeria Syndrome (NGPS), a progeric disease associated with defects in the nuclear envelope (e.g., lamin or baf mutations). This genetic approach, e.g., looking for 'suppressor mutants' of disease phenotypes, is timely and important and provides an inroad for defining drug/drug targets benefiting affected patients in the long term. To the best of my knowledge, I am not in the inner circle of the field; a corresponding knockout mouse has not been described. Thus, the authors cannot expect to provide a mouse model or mouse double mutants as part of their study (showing rescue of progeric phenotypes). I understand that the revision process of this paper is already taking well over a year.

Of course, using *C. elegans* is a very valid nonvertebrate model for aging and a preprint from one or two of the coauthor's labs <https://doi.org/10.1101/2024.03.17.585430> shows that the nematode BAF A12T mutants have a modestly shortened life span compared to wild-type, the phenotype being enhanced in nematodes not carrying a germ line. Besides aging, these worms also show reduced fecundity when aging and other phenotypes are associated with nuclear organization. Thus, it would, in principle, be possible to extend the nematode analysis; this, however, would compromise the novelty of the detailed study described in the preprint. As a reminder, in NCOMMS-23-35016A, the authors use a most valid but maybe more indirect assay to test phenotypes associated with nematode BAF A12T phenotypic suppression: A double mutant with *lamin* and *baf A12T*, which arrests in the L1 stage is used, and the suppression of this phenotype is shown.

All in all, I suggest a two-pronged approach.

1) I think the abstract and possible introduction and discussion need to be further tuned down,

eg. To be more specific in stating that multiple cellular phenotypes associated with progeria could be rescued by Rather than 'identifying 43 new genes that can reverse multiple aging phenotypes in progeria.'

2) I suggest that the editor and key authors have a conversation as to the status of <https://doi.org/10.1101/2024.03.17.585430>. Ideally, I would see a) suppression of direct aging-associated phenotypes (e.g., reduced life span, reduced fecundity, etc.) or b) a better explain the approach of using genetically sensitized nematodes (e.g., *lmn-1* mutant background). This could be easily done by RNAi, but I understand that the authors might be hesitant to compromise the novelty of their study, which may still have to be accepted for publication.

General Note. I suggest not using the 'whole genome synthetic rescue screens', e.g., avoid 'synthetic rescue screen,' which, to my best knowledge, is not used in genetics terminology. E.g., say, a 'genome-wide suppressor screen' a ' a genome-wide screen for genetic suppressors ', 'a genome-wide screen for suppressing and bypassing progeric phenotypes' associated with A12T Guillermo Progeria Syndrome (NGPS) BAF A12T cell line/.

We thank the reviewer for their very insightful comments. We have now modified the language across the manuscript and toned down the claims regarding BAF involvement in protein synthesis (please see track changes across the manuscript). We have also changed "synthetic rescue screen" to "genome-wide suppressor screen" as suggested by the reviewer.

To address the point about the nematode background, please see our response below. We have not incorporated this into the manuscript, following our discussion with the editor.

Rationale for the conducted *C. elegans* experiments

1. Although *baf-1(G12T)* mutants suffer at least two age-related defects, namely reduction in lifespan and reduced brood size, neither of these traits are suitable for a screen of 32 candidate genes. Hence, we developed the idea to use a sensitised background with an easy scorable phenotype.
2. Which background? Genes encoding lamins and LEM-domain proteins were obvious candidates because BAF physically interact with these proteins in the nuclear lamina.
3. Pilot experiments indicated that depletion of LEM-domain protein EMR-1/emerin or LEM-2/LEMD2 by RNAi did not affect viability of *baf-1(G12T)* embryos. In contrast, the combination of *baf-1(G12T)* with *Imn-1(RNAi)* showed a mild increase in lethality compared to *Imn-1(RNAi)* alone.
4. For not having to rely on double RNAi, which potentially suffers reduced efficiency and higher variability, we sought for a suitable *Imn-1* allele.
5. The only available mutant allele of *Imn-1* is *tm1502* (Haithcock et al. 2005). However, *tm1502* is a strong loss-of-function allele (or even a null allele; the first three exons are deleted), which cannot be maintained in homozygosity with the *baf-1(G12T)* allele.
6. Instead, we tested an endogenously tagged *gfp::Imn-1* allele (*yc32*), which was reported to cause approximately 35% embryonic and larval lethality and roughly 50% reduction in fertility (Bone et al. 2016). An independent endogenously tagged *gfp::Imn-1* allele (*jf98*) causes similar lethality (Link et al. 2018). This argues that insertion of *gfp* after the start codon of the *Imn-1* locus generates a partial loss-of-function *Imn-1* allele.
7. Combination of *Imn-1[yc32(gfp::Imn-1)]* with *baf-1(G12T)* leads to a dramatic increase in lethality to almost 100% when nematodes are cultivated on *E. coli* OP50 (Figure S6A) or around 90% with *E. coli* HT115 is used as food source (Figure 7C, Figure S6B).
8. We concluded that this phenotype was suitable for screening of the 32 candidates from the CRISPR genome-wide suppressor screen.

Validation of hits in terms of reversal of *C. elegans* NGPS aging phenotypes

We understand that this would be a relevant addition to our manuscript. However, as the editor and reviewer #4 point out, this would generate a potential conflict with the Romero-Bueno et al manuscript (<https://doi.org/10.1101/2024.03.17.585430>). This manuscript is currently under review at EMBO Journal and as per standard publication policy, we indicated to the EMBO J editor that the data are not under consideration elsewhere.

Under these circumstances, we consider that it would be inappropriate to report in the Nat Comms manuscript that lifespan and fecundity are reduced in *baf-1(G12T)* mutants. In case the Romero-Bueno et al manuscript is accepted for publication, this would open the possibility to perform these validation experiments, but the timeline is obviously unknown.

References:

Bone CR, Chang YT, Cain NE, Murphy SP, Starr DA. 2016. Nuclei migrate through constricted spaces using microtubule motors and actin networks in *C. elegans* hypodermal cells. *Development* **143**: 4193-4202.

- Haithcock E, Dayani Y, Neufeld E, Zahand AJ, Feinstein N, Mattout A, Gruenbaum Y, Liu J. 2005. Age-related changes of nuclear architecture in *Caenorhabditis elegans*. *Proc Natl Acad Sci U S A* **102**: 16690-16695.
- Link J, Paouneskou D, Velkova M, Daryabeigi A, Laos T, Labella S, Barroso C, Pacheco Pinol S, Montoya A, Kramer H et al. 2018. Transient and Partial Nuclear Lamina Disruption Promotes Chromosome Movement in Early Meiotic Prophase. *Dev Cell* **45**: 212-225 e217.

REVIEWERS' COMMENTS

Reviewer #2 (Remarks to the Author):

Breusegem et al. have improved their study in response to the comments of all reviewers. Key questions, however, were not addressed or were not satisfactorily addressed in the revised manuscript. While the authors provided explanations for the relationship between BAF and protein synthesis, the underlying mechanisms have not been definitively addressed. At least, the manuscript should provide an expanded discussion on the role of BAF in regulating protein synthesis, nucleolar size, and the onset of cellular senescence.

Reviewer #4 (Remarks to the Author):

I carefully reread the manuscript, the response by the authors, and the changes they implemented. I support the publication of the manuscript.